JCB Journal of Cell Biology

# Dual protection by Bcp1 and Rkm1 ensures incorporation of uL14 into pre-60S ribosomal subunits

Min-Chi Yeh[1], Ning-Hsiang Hsu[2], Hao-Yu Chu[2], Cheng-Han Yang[1], Pang-Hung Hsu[3,4], Chi-Chi Chou[1], Jing-Ting Shie[2], Wei-Ming Lee[2], Meng-Chiao Ho[1,5,6], and Kai-Yin Lo[2]

**Eukaryotic ribosomal proteins contain extended regions essential for translation coordination. Dedicated chaperones stabilize the associated ribosomal proteins. We identified Bcp1 as the chaperone of uL14 in *Saccharomyces cerevisiae*. Rkm1, the lysine methyltransferase of uL14, forms a ternary complex with Bcp1 and uL14 to protect uL14. Rkm1 is transported with uL14 by importins to the nucleus, and Bcp1 disassembles Rkm1 and importin from uL14 simultaneously in a RanGTP-independent manner. Molecular docking, guided by crosslinking mass spectrometry and validated by a low-resolution cryo-EM map, reveals the correlation between Bcp1, Rkm1, and uL14, demonstrating the protection model. In addition, the ternary complex also serves as a surveillance point, whereas incorrect uL14 is retained on Rkm1 and prevented from loading to the pre-60S ribosomal subunits. This study reveals the molecular mechanism of how uL14 is protected and quality checked by serial steps to ensure its safe delivery from the cytoplasm until its incorporation into the 60S ribosomal subunit.**

## Introduction

Proteins play crucial roles in various physiological functions, such as forming cell structures, mediating cell signaling, and providing energy. Ribosomes are highly conserved ribonucleoprotein complexes in protein translation. In eukaryotic and prokaryotic cells, ribosomes consist of two subunits: the 60S and 40S, or 50S and 30S, respectively. In yeast, the 40S subunit comprises 18S rRNA and 33 ribosomal proteins, while the 60S subunit comprises 25S, 5.8S, and 5S rRNAs, and 46 ribosomal proteins. Transcription of 25S, 18S, and 5.8S rRNAs is carried out by RNA polymerase I, while 5S rRNA is transcribed separately by RNA polymerase III. Following RNA synthesis, ribosomal proteins and transacting factors are loaded to form preribosomal subunits. Over 200 transacting factors participate in this process in eukaryotes, facilitating the processing and editing of nascent rRNAs. These factors also ensure a hierarchical assembly process with quality control at each assembly step. Due to the high negative charges of ribosomal subunits and the requirement of binding partners to be stabilized, numerous factors are required for their nuclear export (see reviews Baßler and Hurt, 2019; Dörner et al., 2023; Konikkat and Woolford, 2017; Kressler et al., 2017; Peña et al., 2017).

Ribosomal proteins play an important role in the intricate process of ribosome assembly, providing a positively charged surface for rRNA binding that accelerates the maturation of the ribosome (Jomaa et al., 2014; Pecoraro et al., 2021). Throughout evolution, ribosomal proteins have developed unstructured regions that enable communication between different domains of the ribosome. However, these characteristics make ribosomal proteins susceptible to degradation or aggregation prior to their incorporation into the ribosome. To counteract these challenges, a combination of general chaperone systems, karyopherin, and dedicated chaperons is required to ensure the stability of ribosomal proteins (Pillet et al., 2017).

Many dedicated chaperones for ribosomal proteins are identified and are crucial for stabilizing specific ribosomal proteins (Calviño et al., 2015; Eisinger et al., 1997; Holzer et al., 2013; Iouk et al., 2001; Koch et al., 2012; Kressler et al., 2012; Pausch et al., 2015; Pillet et al., 2015; Schütz et al., 2014; Stelter et al., 2015; West et al., 2005). These chaperones have additional functional roles for the assembly of the ribosomal proteins. They may facilitate the import process, ensuring the incorporation of ribosomal proteins into nascent ribosomes in the nucleus (Kressler et al., 2012; Pillet et al., 2015; Stelter et al., 2015).

[1]Institute of Biological Chemistry, Academia Sinica, Taipei, Taiwan; [2]Department of Agricultural Chemistry, College of Bioresources and Agriculture, National Taiwan University, Taipei, Taiwan; [3]Department of Bioscience and Biotechnology, College of Life Science, National Taiwan Ocean University, Keelung, Taiwan; [4]Center of Excellence for the Oceans, National Taiwan Ocean University, Keelung, Taiwan; [5]Institute of Biochemical Sciences, College of Life Science, National Taiwan University, Taipei, Taiwan; [6]Institute of Biochemistry and Molecular Biology, College of Medicine, National Taiwan University, Taipei, Taiwan.

Correspondence to Kai-Yin Lo: kaiyin@ntu.edu.tw; Meng-Chiao Ho: joeho@gate.sinica.edu.tw.

**Rockefeller University Press**
J. Cell Biol. 2024 Vol. 223 No. 8 e202306117



Symportin 1 imports uL18 (Rpl5) and uL5 (Rpl11) with correct stoichiometry and chaperones 5S RNP assembly (Bange et al., 2013; Calviño et al., 2015; Kressler et al., 2012). A recent study shows that eS26 (Rps26) and uL16 (Rpl10) are preferentially oxidized, and the non-functional ribosomal proteins could be replaced by their dedicated chaperones, Tsr2 and Sqt1, from the mature ribosomal subunits. This chaperone-mediated ribosome repair is essential for oxidative stress resistance, correlating with aging and health (Yang et al., 2023). The transportation of ribosomal proteins into the nucleus involves direct interaction with karyopherins, and their release is typically mediated by a RanGTP-dependent mechanism. Nevertheless, nuclear chaperones can also play a role in facilitating the release of ribosomal proteins, employing a RanGTP-independent process (Schütz et al., 2014; Ting et al., 2017).

uL14 is a conserved ribosomal protein found in prokaryotes and eukaryotes. It is encoded by *RPL23A* and *RPL23B* genes in yeast. Positioned at the center of the interface between the 40S and 60S subunits, uL14 plays a crucial role by serving as the primary binding site for Tif6 (Klinge et al., 2011). Tif6, the yeast homolog of eIF6 (initiation factor 6), is a transacting factor of 60S biogenesis. It actively participates in biogenesis, starting from the early stages of rRNA processing (Basu et al., 2003) and continuing through to almost the final maturation step (Lo et al., 2010). The binding of Tif6 prevents the premature association between non-matured 60S and 40S subunits. The importance of this mechanism is underscored by the connection to Shwachman–Diamond syndrome. In instances where there is a mutation affecting its release factor, Sdo1, Tif6 tends to be retained on the 60S subunits, leading to an inadequacy of mature 60S subunits (Gijsbers et al., 2013; Menne et al., 2007).

Bcp1 exports Mss4 (phosphoinositol-4-kinase) from the nucleus, whereas Mss4 can synthesize phosphoinositol at the plasma membrane (Audhya and Emr, 2003). Our previous study found that Bcp1 acts as a nuclear chaperone of uL14 (Rpl23) (uL14 in the new nomenclature (Jenner et al., 2012)), dissociating uL14 from the importins and maintaining its stability (Ting et al., 2017). The human homolog, BCCIP (BRCA2 and p21 interacting protein), can also stabilize nuclear uL14 (Wyler et al., 2014) and is required for nucleolar recruitment of eIF6 and 12S pre-rRNA production during 60S ribosome biogenesis (Ye et al., 2020).

Tsr2 and Bcp1 are also identified as escortins. They can release ribosomal proteins from importins independently of RanGTP and deliver them safely to the assembly site on nascent ribosomal subunits (Schütz et al., 2014; Ting et al., 2017). The previous study showed that the interaction between Tsr2 and the eukaryotic-specific segments (ESS) in the eS26 is required to prompt a non-canonical RanGTP-independent disassembly of eS26 from importins. The deletion of the ESS of eS26 maintains its interaction with importins but prevents its release by Tsr2 (Schütz et al., 2018). However, how Bcp1 protects and releases uL14 from the importins is unknown.

Here, we found that Rkm1 also plays a role in stabilizing uL14. Rkm1, identified as a lysine methyltransferase, possesses a conserved SET (suppressor of variegation, enhancer of zeste, and trithorax) domain in yeast. It is identified as a methyltransferase for uL14, responsible for dimethylations at lysine 105 and 109 (Porras-Yakushi et al., 2005, 2007). Additionally, Rkm1 is involved in the monomethylation of lysine 48 in the 40S ribosomal protein uS13 (Rps18) (Couttas et al., 2012). In yeast, there are 10 methyltransferases for ribosomal proteins, and they collectively contribute to various aspects of ribosome biology, including ribosome biogenesis, translation elongation fidelity, and translation termination (Al-Hadid et al., 2016a, 2016b). Despite its involvement in these processes, the loss of Rkm1 does not significantly impact cellular growth (Porras-Yakushi et al., 2007). However, it does lead to a subtle under-accumulation of 60S subunits and a minor decrease in translation fidelity (Al-Hadid et al., 2016b). Nevertheless, the precise physiological role of Rkm1 remains unclear, warranting further exploration and understanding.

This study elucidated the structure model of the ternary complex involving Bcp1, uL14, and Rkm1, and systematically analyzed the protective mechanisms. In the absence of Rkm1, cells exhibited a growth rate comparable with the wild type. However, the deletion of *RKM1* in *bcp1ts* cells not only impeded cell growth but also exacerbated the reduction in nascent uL14 levels. This finding underscores the synergistic role of Rkm1 in uL14 protection. Rkm1 accompanies uL14 during the transport by importins. Upon the association of Bcp1, there is a cascade effect triggering the release of importin and Rkm1 from uL14 in the nucleus. Bcp1 assumes the protective role at this stage, safeguarding uL14 until its delivery to the pre-60S subunit. Molecular docking, guided by XL-MS and validated through cryo-EM, reveals that both Bcp1 and Rkm1 play a crucial role in protecting the internal loop of uL14, a region vital for 60S incorporation. Notably, mutant uL14, when present, tends to be retained in the ternary complex, preventing its successful incorporation into the pre-60S subunit. This sequential protection mechanism extends beyond mere safeguarding; it also functions as a surveillance system for ribosomal proteins, ensuring their integrity throughout the intricate assembly process.

## Results

### Bcp1 and Rkm1 form a complex with uL14 to maintain the stability of uL14

Bcp1 is a nuclear chaperone of uL14, demonstrating its ability to dissociate uL14 from karyopherins and stabilize uL14 through direct protein–protein interactions (Ting et al., 2017). Rkm1 is a SET domain methyltransferase for uL14, responsible for the dimethylation of Lys 106 and Lys110 (Porras-Yakushi et al., 2005). A large-scale yeast-two-hybrid assay revealed an interaction between Bcp1 and Rkm1 (Yu et al., 2008). To explore the potential connection further, we examined the genetic interaction between Rkm1 and Bcp1 using a *bcp1ts* temperature-sensitive mutant, which exhibits slow growth at higher temperatures (33 and 35°C) (Fig. 1 A). While *rkm1Δ* did not show growth defect at the normal or stress conditions tested (Fig. S1 A), the deletion of *RKM1* in *bcp1ts* resulted in a more severe growth defect (Fig. 1 A), indicating a functional interdependence.

Further investigations delved into the in vitro interactions among Bcp1, Rkm1, and uL14. Examination by the size exclusion chromatography revealed that both Bcp1 and Rkm1 coeluted

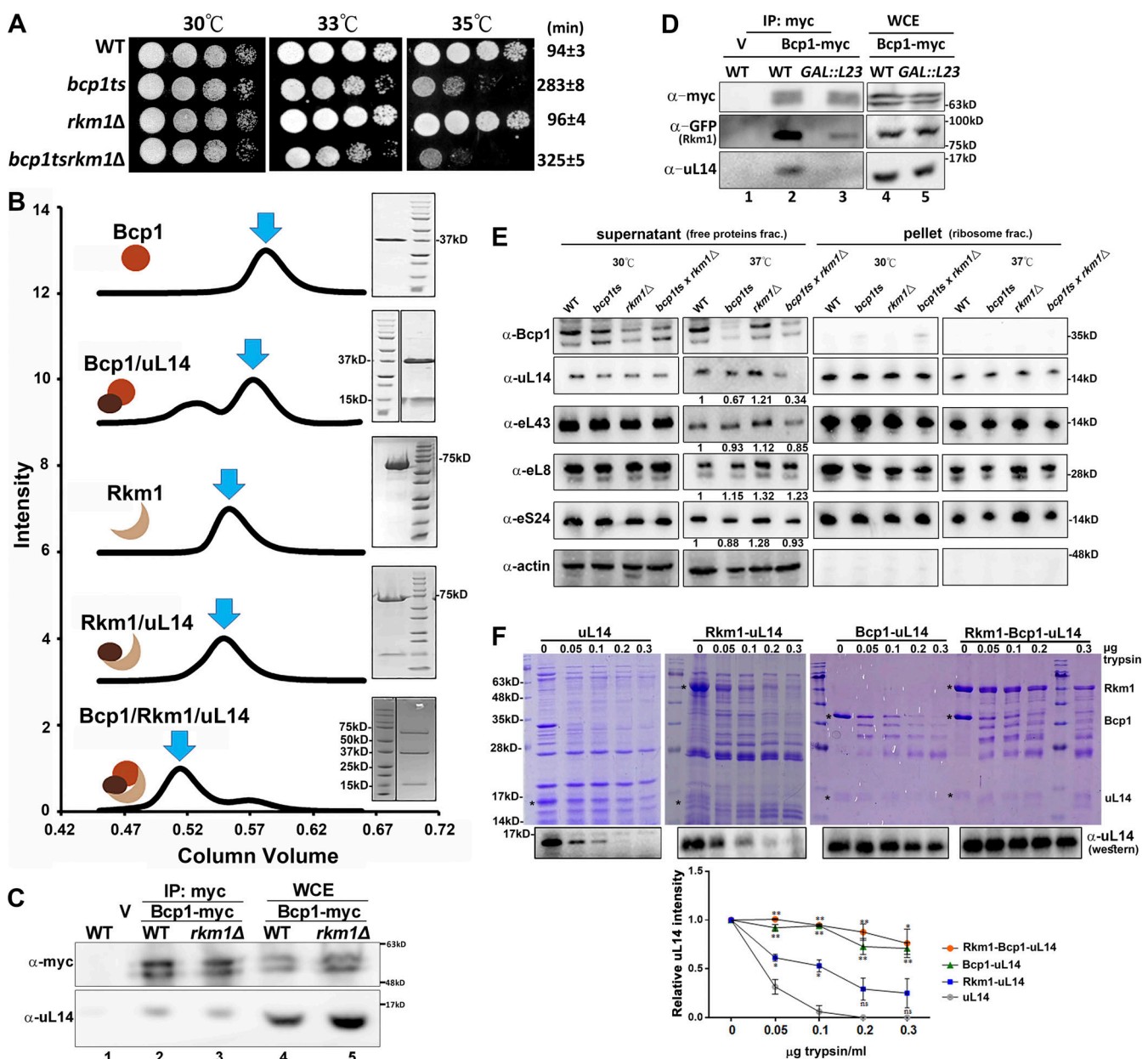

Figure 1. **Bcp1 and Rkm1 form a complex with uL14 and maintain the stability of uL14. (A)** Wild-type, *bcp1ts*, *rkm1Δ*, and *bcp1tsrkm1Δ* were normalized and spotted on the YPD plates. The plates were incubated at the temperatures indicated in the figure for 2–3 days. The doubling time was estimated from three independent samples cultured in liquid YPD medium at 35°C (Ave ± SD). **(B)** The size exclusion chromatography (left). The result of Coomassie blue gels (right). The y-axis is the normalized intensity of UV280 nm. The x-axis is the column volume of the Superdex 200 column. The peaks indicated with blue arrows were collected and analyzed in SDS-PAGE. **(C and D)** Bcp1-myc was immunoprecipitated from wild-type and *rkm1Δ* (C). Bcp1-myc was immunoprecipitated from wild-type and *GAL::RPL23* strain. Cells were cultured in Leu⁻ medium containing 2% galactose to OD 0.2–0.3, and 2% glucose was added for another 4 h (D). The associated proteins were detected by Western blotting. **(E)** The normalized cell lysates were prepared from the strains above and spun at 80,000 rpm for 1 h to separate free proteins (supernatant) and ribosomal subunits (pellet). The supernatants were precipitated with TCA and analyzed by Western blotting. **(F)** Various amounts of trypsin, as indicated in the figure, were added to purified uL14 or the purified complexes of Rkm1/uL14, Bcp1/uL14, and Rkm1/Bcp1/uL14, followed by incubation at 37°C for 30 min. The samples were subsequently analyzed using SDS-PAGE with Coomassie blue staining or western blotting. The intensity of uL14 was quantified using Image J, and the relative amounts were calculated compared with the control (no trypsin). Three independent replicates were conducted, and the Student's *t* test was performed against uL14 alone to assess statistical significance (*P < 0.05; **P < 0.01). Source data are available for this figure: SourceData F1.

with uL14, indicating the formation of stable uL14/Bcp1 and uL14/Rkm1 complexes. In addition, Bcp1, uL14, and Rkm1 also formed a stable complex (Fig. 1 B). The protein band intensity in SDS-PAGE and molecular weight estimation from size exclusion

chromatography suggested an approximate 1:1:1 ratio for each protein component in the ternary complex. However, Bcp1 and Rkm1 did not coelute in the absence of uL14 (Fig. S1 B), implying that uL14 acts as a necessary bridge between Bcp1 and Rkm1 in

the formation of this complex, highlighting the intricate nature of their interactions.

To further elucidate the in vivo formation of the ternary complex, we conducted immunoprecipitation assays in yeast. Given that *RKM1* is a nonessential gene, a deletion mutant of *RKM1* (*rkm1Δ*) was used. Since *RPL23* is an essential gene, we utilized *GAL::RPL23*, a conditional mutant constructed under the glucose-repressible *GAL10* promoter. Bcp1-myc was immunoprecipitated with anti-myc antibodies and protein-A beads. Consistent with the in vitro interaction results, uL14 and Rkm1 coimmunoprecipitated with Bcp1 (Fig. 1, C and D). In the *rkm1Δ* mutant, the interaction between Bcp1 and uL14 remained unchanged (Fig. 1 C, lane 3). Conversely, the interaction between Bcp1 and Rkm1 decreased when uL14 was depleted (Fig. 1 D, lane 3). Although the total level of uL14 did not significantly decrease because most proteins are incorporated into the 60S subunits, which are very stable complexes (Fig. 1 D, lane 5), the nascent uL14 rapidly declined upon the glucose-induced shutdown of the *GAL10* promoter. These data suggest that the in vivo formation of the hetero-trimeric complex involving Bcp1, Rkm1, and uL14 is sensitive to the nascent uL14 level.

In our previous investigation, we identified uL14 as a dosage suppressor of the *bcp1ts* mutant, establishing Bcp1 as the chaperone for uL14. The overexpression of Bcp1 was found to stabilize nascent uL14, whereas the inactivation of Bcp1 destabilized nascent uL14 (Ting et al., 2017). Given that Rkm1 is an interaction partner of Bcp1, we explored whether overexpression of *RKM1* could similarly rescue the growth defect of *bcp1* mutant. However, high-copy *RKM1* did not alleviate the growth defects of the *bcp1ts* mutant at 37°C (Fig. S1 C). In addition, the deletion of *RKM1* did not result in a decrease in the protein level of Bcp1 (Fig. S1 D). Moreover, alterations in Bcp1 levels did not affect the abundance of Rkm1 (Fig. S1 E).

Posttranslational modifications are important for protein–protein interactions, functional regulation, and protein stability. Since the absence of Rkm1 exacerbated the growth of *bcp1ts* (Fig. 1 A), the nascent uL14 level was examined under this condition. To focus on nascent ribosomal proteins, which exist in limited amounts before incorporation into pre-ribosomes, we separated free proteins and ribosomal subunits through ultracentrifugation of whole-cell lysates. The level of free uL14 decreased in *bcp1ts* at 37°C (Fig. 1 E) compared with the wild-type, aligning with the known chaperone function of Bcp1 (Ting et al., 2017). In contrast, the nascent uL14 signal remained unaffected in the *rkm1Δ* mutant, but it was nearly undetectable in the *bcp1tsrkm1Δ* mutant at 37°C. Other ribosomal proteins were detected for comparisons, and it was observed that they did not exhibit a significant decrease comparable with uL14 under the same conditions (Fig. 1 E).

To further substantiate the protective role of Bcp1 and Rkm1 for uL14, we conducted limited proteolysis experiments. Varying amounts of trypsin were introduced to purified uL14, as well as to the purified complexes of Rkm1/uL14, Bcp1/uL14, and Rkm1/Bcp1/uL14. The stability of uL14 was then compared across these conditions. uL14 exhibited the highest sensitivity to trypsin, but the addition of Rkm1 or Bcp1 conferred protection against proteolysis, with Bcp1 demonstrating superior protective capability

compared with Rkm1. Remarkably, the presence of both Bcp1 and Rkm1 significantly enhanced the stability of uL14 (Fig. 1 F). These findings underscore the essential role of the collaborative action between Bcp1 and Rkm1 in stabilizing nascent uL14.

## Rkm1 accompanies the transport of uL14 to the nucleus

Ribosomal proteins are translated in the cytoplasm and imported into the nucleus for ribosome assembly. We previously showed that Bcp1 interacts with uL14 in the nucleus (Ting et al., 2017). Here, we asked at what stage Rkm1 binds uL14. Rkm1-GFP showed both cytoplasmic and nuclear signals in wild-type cells. In the NLS prediction (Ba et al., 2009), Rkm1 contains no NLS and uL14 contains an NLS. This makes us wonder if Rkm1 depends on uL14 for import. We tracked Rkm1-GFP in the *GAL-RPL23* strain, whereas the expression of uL14 was under the control of a *GAL*-driven promoter. In galactose condition, Rkm1 was present in the nucleus in WT and *GAL-RPL23* strains. However, it lost nuclear signal upon depletion of uL14 (Fig. 2 A, +Glc). The potential interdependence was also examined, and it was found that Bcp1 remained in the nucleus under the depletion of uL14, while Rkm1 remained in the nucleus in the *bcp1ts* strain (Fig. S1 F). These data suggest a dependence on uL14 in the Rkm1 transport pathway.

Nuclear import of ribosomal proteins majorly depends on the importins Kap121 and Kap123 (Rout et al., 1997). Recombinant Kaps were expressed as N-terminal glutathione transferase (GST) fusion proteins and incubated with Rkm1, uL14, or Rkm1/uL14 complex in vitro. Both Kap121 and Kap123 interacted with uL14 directly (Fig. 2 B, lanes 4 and 8) but barely interacted with Rkm1 (Fig. 2 B, lanes 5 and 9). Interestingly, Rkm1 binding was enhanced by uL14 in Kap121 and Kap123 (Fig. 2 B, lanes 6 and 10).

To further demonstrate that Rkm1 is a shuttling protein, its cellular distribution was examined in the *kap121ts*, *kap123Δ*, and *crm1T539C* mutants. The nuclear signals of Rkm1 became diffused in the cytoplasm in *kap123Δ*, with a less significant change observed in *kap121ts* mutants (Fig. 2 C). Crm1 is the primary export pathway in cells (Stade et al., 1997). In *crm1T539C*, which is sensitive to the inhibitor leptomycin B, Nmd3-GFP was included as a control. Consistent with a previous study (Ho et al., 2000), Nmd3-GFP was trapped in the nucleus in the presence of leptomycin B. Compared with WT, Rkm1 became more concentrated in the nucleus when the Crm1 export pathway was inactivated (Fig. 2 D). The above data suggest that the import of Rkm1 depends on uL14 via Kap123 and is exported via Crm1 after its release from uL14.

Bcp1 acts as an escortin, facilitating the dissociation of uL14 from karyopherins in a Ran-GTP-independent manner (Ting et al., 2017). Given this role, it is plausible that Bcp1 also plays a similar role in dissociating the Rkm1-uL14 complex from karyopherin (Fig. 2 E). To test this hypothesis, we immobilized complexes of karyopherin, Rkm1, and uL14 on beads and introduced purified Bcp1 for the release test. In alignment with our previous observation (Ting et al., 2017), Bcp1 successfully released Rkm1 and uL14 simultaneously from Kap121 and Kap123 (Fig. 2 E).

To investigate the association of Rkm1 with the 60S subunit, we detected the distribution of Rkm1 across sucrose gradients.

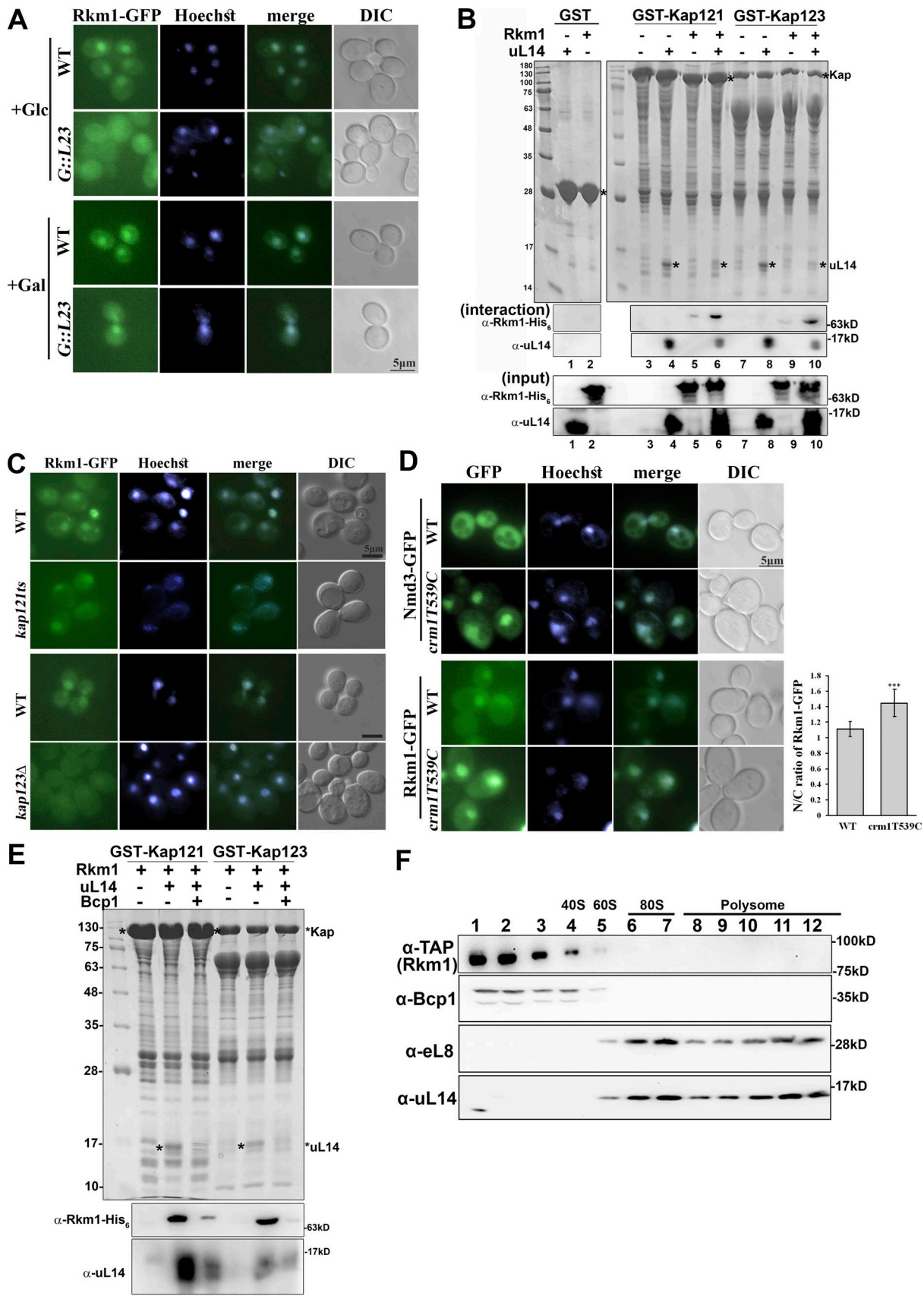

Figure 2. **Rkm1 accompanies the transport of Rpl23 to the nucleus. (A)** The localization of Rkm1-GFP was visualized in wild-type and *GAL::RPL23*. Overnight cells were subcultured in a medium containing 2% galactose for 2 h. Cultures were kept in the galactose, or 2% glucose was added for another 4 h

before examination with fluorescence microscopy. **(B)** GST-Kap121 or GST-Kap123 was incubated with uL14, Rkm1, or both at 4°C for 1 h. After three times of wash, the proteins were eluted with 1X SDS sample buffer and analyzed by Coomassie blue staining and western blotting. **(C)** To visualize the localization of Rkm1-GFP, wild-type and *kap121ts* at log phase were shifted to 37°C for 2 h, and wild-type and *kap123Δ* were cultured at 30°C. **(D)** Nmd3-GFP and Rkm1-GFP localizations were monitored in wild-type and crm1T539C incubated with LMB (0.1 μg/ml) for 30 min. Hoechst was used to stain the nucleus. The intensity ratios between nucleus and cytoplasm were calculated in 15 cells and analyzed with Student's *t* test (***P < 0.001). **(E)** GST-Kap121/Kap123 in complex with Rkm1 or Rkm1/uL14 was immobilized on the glutathione beads. Bcp1 was added and incubated for another hour at 4°C. The remaining amounts of Rkm1 and uL14 on the beads were analyzed by Western blotting. **(F)** Cell extracts from Rkm1-TAP were fractioned through 7–47% sucrose gradients. Each fraction was precipitated with TCA and analyzed by anti-TAP, anti-Bcp1, anti-eL8, and anti-uL14 antibodies. The corresponding sedimentation peaks of the ribosomal subunits were indicated above. Source data are available for this figure: SourceData F2.

---

Cells expressing Rkm1-TAP were grown to the early log phase, and the cell extracts were fractioned through a 7–47% linear sucrose gradient. The majority of Rkm1 was observed at the top of the gradients, similar to the distribution pattern of Bcp1 (Fig. 2 F). This result suggests that Rkm1 and uL14 interact in the cytoplasm and are cotransported by karyopherins. Bcp1 then facilitates the dissociation of Rkm1 and uL14 from Kaps, forming a heterotrimeric complex. However, it is noteworthy that this complex is not associated with the 60S subunits, or alternatively, it interacts with the 60S in a dynamic manner (Fig. 2 F).

### The structure of Bcp1, Rkm1, and uL14 ternary complex

Attempts to cocrystallize the Bcp1/ul14 heterodimer, Rkm1/ul14 heterodimer, and Bcp1/ul14/Rmk1 ternary complex proved unsuccessful. Due to the structural heterogeneity, cryo-electron microscopy (cryo-EM) could only provide a low-resolution molecular envelope of the uL14/Bcp1/Rkm1 ternary complex, forming a distinctive heart-shaped structure (Fig. S2).

To elucidate the interaction among three proteins, we applied chemical crosslinking coupled with mass spectrometry (XL-MS) to identify proximate amino acid residues and interaction relationships within the ternary complex (Herzog et al., 2012). Glutaraldehyde, $CH_2(CH_2CHO)_2$, was used to crosslink two ε-amino groups of lysine side chains (Bishop and Richards, 1968). To ensure specificity in crosslinking, we isolated the ternary complex with the correct molecular weight through SDS-PAGE electrophoresis prior to mass spectrometry analysis (Migneault et al., 2004). Our XL-MS results showed two pairs of intermolecular crosslinks between Bcp1 and Rkm1, one pair of intermolecular crosslinking between Bcp1 and uL14, and two pairs of intermolecular crosslinking between Rkm1 and ul14 (Fig. 3 A; and Figs. S3, S4, S5, S6, and S7).

Currently, the apo-Bcp1 structure has been determined (Lin et al., 2020), and the uL14 structure can be extracted from pre-60S ribosome structures (PDB: 5H4P) (Ma et al., 2017). The structural model for Rkm1 has been predicted by Alphafold with very high confidence (Jumper et al., 2021). The predicted Rkm1 model shows an elongated shape, featuring an N-terminal conserved SET domain (residue 1–288) and a C-terminal domain rich in alpha-helices (Fig. S4). Leveraging information from XL-MS, we generated a crosslink guided molecular model of Bcp1/uL14/Rkm1 ternary complex using the reported protocol (Fig. 3 B) (Kahraman et al., 2013). This docking model aligns reasonably well with the heart-shaped envelope derived independently from our cryo-EM experiment, supporting our docking model (Fig. 3 C).

In our model, the ESS2 region (loop[42-50]) of uL14, which is buried within the ribosome for interaction with 25S rRNAs, is protected by Bcp1 and Rkm1 (Fig. 4 A). This finding is consistent with previous research suggesting that chaperons protect ribosomal proteins by shielding their positively charged regions interacting with rRNAs before incorporation into the ribosome (Pillet et al., 2017). Deletion of this loop resulted in lethality (Fig. 4 B), abolished the interaction with Bcp1 (Fig. 4 C), but maintained the interaction with Rkm1 (Fig. 4 D). However, the mutation of Arg45 and Arg48, crucial residues for ribosome interaction, to alanine did not affect growth and Bcp1 interaction (Fig. 4, B and C, RA mutant). The immunoprecipitation data consistently demonstrated that uL14(Δloop) lost interaction with Bcp1 but not with Tif6 and Rkm1 in vivo (Fig. 4 E).

Furthermore, our structural analysis indicates that uL14 is sandwiched between Rkm1 and Bcp1. Specifically, uL14 is positioned in close proximity to the SET domain of Rkm1 for lysine methylation (Fig. 4 F). Molecular dynamics simulation shows that protein dynamics enable the movement of Lys106 and Lys110 residues to the Rkm1 active site for methylation (Fig. 4 G and Video 1).

### The interaction between Bcp1 and uL14 is essential to release uL14 from Kap

The 36 amino acids at the N terminus of Bcp1 are missing in the X-ray structure, and the sequence from amino acids 37–52 is unstructured (Lin et al., 2020) (Fig. 5 A). The N terminus contains a DE-rich sequence and a potential monopartite nuclear localization signal (NLS) at amino acids 13–16, as predicted by NLS prediction programs (Psort II and NLS mapper) (Fig. 5 B). To demonstrate its function as NLS, mutants *bcp1(ΔN10)*, *bcp1(ΔN20)*, and *bcp1(ΔN40)* were generated by serial truncations of 10, 20, and 40 amino acids from the N terminus. While *bcp1(ΔN10)* could support the growth of the *bcp1ts* mutant, *bcp1(ΔN20)* partially complemented the growth, and *bcp1(ΔN40)* could not support the growth (Fig. 5 C).

The cellular localizations of these mutants were monitored. Wild-type Bcp1 was predominantly localized in the nucleus, whereas bcp1(ΔN10) showed nuclear and enhanced cytoplasmic signal, and bcp1(ΔN20) and bcp1(ΔN40) were mislocalized to the cytoplasm (Fig. 5 D). Consistently, these mutants displayed decreased interaction with importins. Recombinant bcp1ΔN10, bcp1ΔN20, and bcp1ΔN40 proteins were overexpressed in *E. coli* and utilized in interaction studies. GST-Kap121 was applied to test the interaction with various N-terminal truncated Bcp1 mutants. While full-length Bcp1 interacted with Kap121, bcp1ΔN10 exhibited reduced interaction, and bcp1ΔN20 and

**Figure 3. The predicted model of Bcp1/uL14/Rkm1 ternary complex. (A)** XL-MS reveals crosslinks between Bcp1, uL14, and Rkm1. NLS: nuclear localization signal. ESS: eukaryotic-specific segments. CC: Coiled-coils. **(B)** The ternary complex of Bcp1 (green), Rkm1 (pink), and uL14 (blue) calculated based on XL-MS and cryo-EM map. The crosslinking residues between two molecules of the ternary complex are shown. **(C)** The cryo-EM map of the Bcp1/uL14/Rkm1 complex.

bcp1ΔN40 lost the interaction with Kap121 (Fig. 5 E, glutathione beads). In contrast, all N-terminal-deletion Bcp1 variants maintained interaction with uL14 (Fig. 5 E, NTA beads). The disruption with karyopherin was specific but not due to structural alterations.

To dissect how Bcp1 releases uL14 from Kap, Bcp1 and uL14 mutants were included in the release study. The purified GST-Kap121 and uL14 complex were immobilized on the glutathione beads. Although bcp1ΔN10 and bcp1ΔN20 exhibited reduced interaction with Kap121 (Fig. 5 E), they showed a similar release ability as full-length Bcp1 (Fig. 5 F, lanes 3–5). In contrast, while uL14Δloop maintained its interaction with Kap121, it was barely released by Bcp1 (Fig. 5 F, lanes 7–9).

We further examined this connection in vivo. Tap-tagged Kap121/Kap123 were immunopurified in the WT with additional expressions of uL14 or uL14Δloop. In the presence of the uL14Δloop, the interactions with Rkm1 were enhanced in Kap123 but not significantly in Kap121 (Fig. 5 G). Thus, the interaction between Bcp1 and uL14 is crucial for releasing the uL14/Rkm1 complex from karyopherin.

**Bcp1 triggers the release of Rkm1 from methylated uL14**

The data presented above suggest that Rkm1 and uL14 initially interact in the cytoplasm and are cotransported into the nucleus. The prolonged association between Rkm1 and uL14 is intriguing,

as conventional enzymatic logic implies that an enzyme should release its substrate once the catalysis is complete. Two possible explanations arise: either Rkm1 remains inactive until reaching the nucleus or an additional factor is required for substrate release.

To explore these possibilities, we conducted an in vitro methylation assay. The purified Rkm1/uL14 complex was immobilized on glutathione beads (Fig. 6 A). No methylation was detected in the absence of S-adenosyl methionine (SAM) (Fig. 6 A, lane 1). When SAM was introduced as a methyl donor, uL14 underwent methylation while remaining bound to Rkm1 (Fig. 6 A, lane 2). The addition of Bcp1 resulted in the formation of a ternary complex. Notably, the inclusion of both Bcp1 and SAM led to the release of a significant portion of methylated uL14 from Rkm1 (Fig. 6 A, lanes 4 and 8).

To validate the specificity of uL14 methylation by Rkm1, we generated a catalytically inactive mutant of Rkm1. Through sequence alignment across Rkm1 and other SET methyltransferases, we pinpointed Y273 as a crucial residue for the transfer of a methyl group from SAM to the substrate (Fig. S8 A). Compared with the SETD6 structure (Chang et al., 2011), this site is at the catalytic center of the SET domain (Yeates, 2002) on the Rkm1 structure (Fig. S8 B). This tyrosine was mutated to phenylalanine, generating the rkm1(Y273F) mutant. The Y273F mutation in Rkm1 abrogated uL14 methylation while

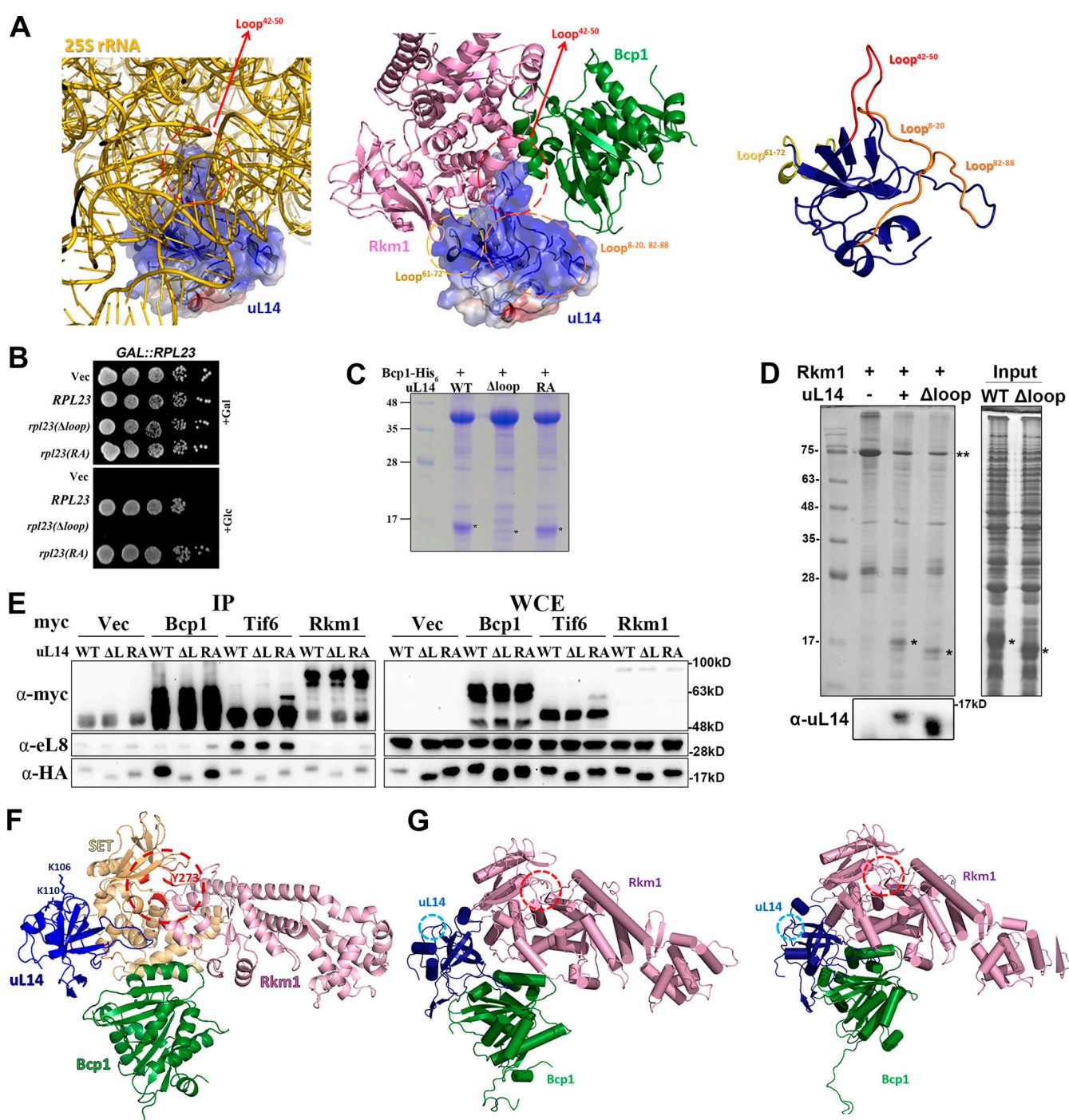

Figure 4. **The positively charged region (loop 42–50) of uL14 that is buried into the ribosome for interacting with 25S rRNAs is protected by Bcp1 and Rkm1. (A)** Compare the uL14 interaction within the pre-60S and Rkm1/Bcp1 complex. Left: uL14 was shown in 60S subunits, and the surrounded 25S rRNA was shown in gray. PDB accession no. 5H4P. Middle: uL14 was shown with Bcp1 and Rkm1. Right: The extension loops of uL14 were shown. **(B)** The growth complementation assays of the *uL14Δloop* and *uL14(RA)*. **(C)** The interactions between recombinant Bcp1 and uL14 mutants were tested with Ni-NTA beads. **(D)** The interactions between Rkm1 and uL14 and uL14(Δloop) were examined in vitro. **(E)** Bcp1-myc, Tif6-myc, and Rkm1-myc were immunoprecipitated in the wild-type strain using protein A-coupled beads. The associated uL14 mutants were examined with the anti-HA antibody. Anti-eL8 antibody was used to probe 60S subunits. **(F)** The SET domain of Rkm1 was labeled in yellow, and the residues at the active site (red circle) were labeled in red. K106 and K110 were also shown on the uL14. **(G)** The model relaxation by molecular dynamics simulation shows the dynamic movement of this ternary complex at two states. The Lys106 and Lys110 residues of uL14 (blue circle) move to the Rkm1 active site (red circle) for methylation. Rkm1: pink; Bcp1: green; uL14: blue. (Please see Video 1 for the animation). Source data are available for this figure: SourceData F4.

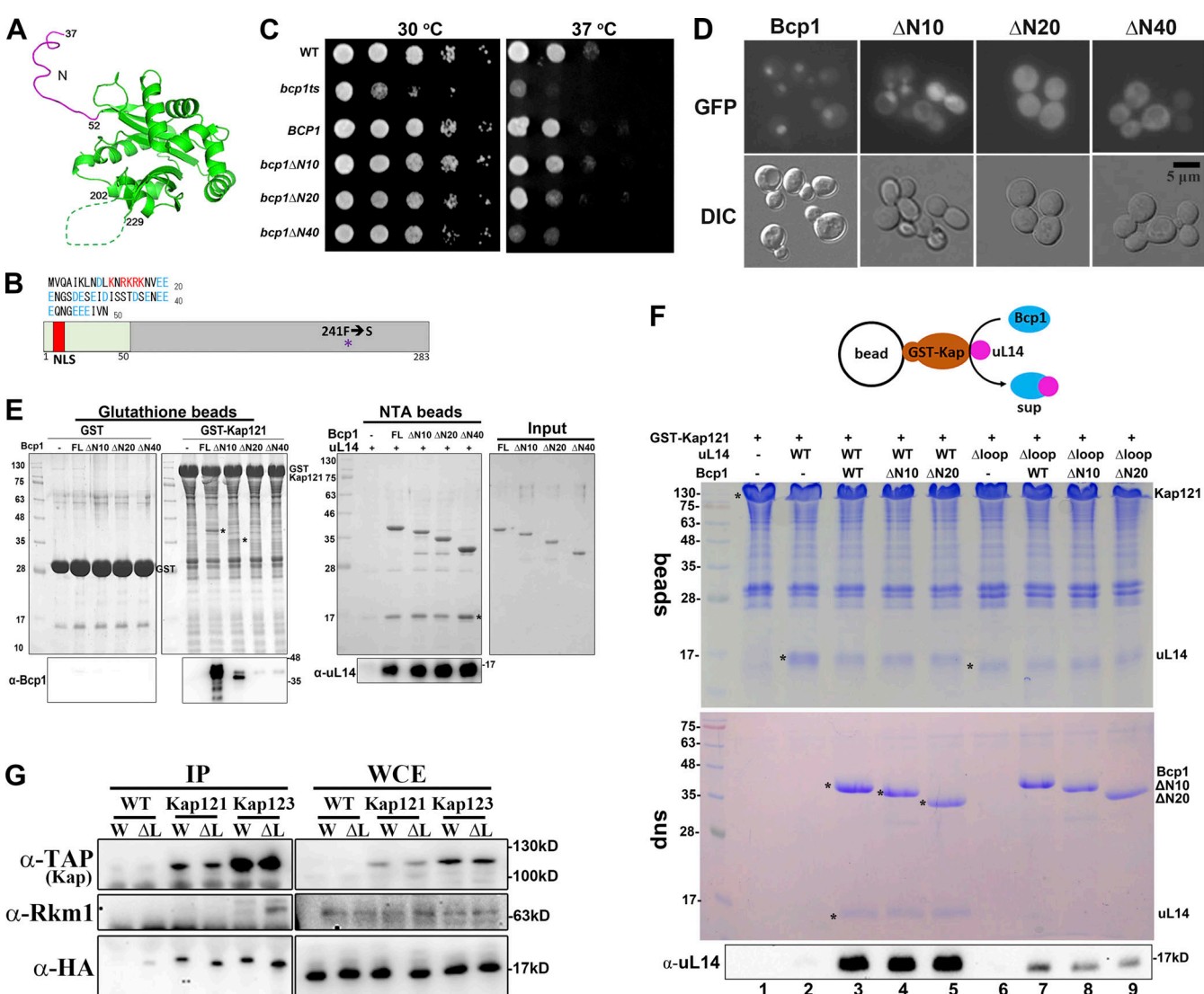

Figure 5. **The interaction between Bcp1 and uL14 is important to release uL14 from Kap. (A)** The N-terminus (magenta) is shown on the Bcp1 structure. **(B)** The diagram of Bcp1 shows the N terminus and NLS sequences. The mutation site of *bcp1ts* is F241S. **(C)** *BCP1*, *bcp1ΔN10*, *bcp1ΔN20*, and *bcp1ΔN40* were transformed to *bcp1ts* and applied in the growth test. **(D)** The localization of Bcp1-GFP, bcp1ΔN10-GFP, bcp1ΔN20-GFP, and bcp1ΔN40-GFP were examined under fluorescence microscopy. **(E)** GST-Kap121 was immobilized on the glutathione beads and interacted with purified Bcp1, bcp1ΔN10, bcp1ΔN20, and bcp1ΔN40 (left panel). The positions of Bcp1 were indicated with asterisk (*), and the interaction signals were also detected with α-Bcp1 antibody. In parallel, bcp1ΔN mutants were immobilized on the NTA beads, and the interactions with uL14 were also examined (middle panel). The position of uL14 was indicated with asterisk (*) and the interaction signals were also detected with α-uL14 antibody. The purified Bcp1 and bcp1ΔN mutant proteins were shown (right panel, input). **(F)** GST-Kap121 in complex with uL14 or uL14Δloop was immobilized on the glutathione beads. Purified Bcp1, bcp1ΔN10, or bcp1ΔN20 was added and incubated for another hour at 4°C. The partitions of uL14 on the beads and in the supernatants were analyzed. The uL14 signals in the supernatants were detected by western blotting. **(G)** Kap121-TAP and Kap123-TAP were immunopurified in WT with additional expressions of uL14 or uL14Δloop. The associated proteins were probed with anti-Rkm1 and anti-HA antibodies. Source data are available for this figure: SourceData F5.

preserving the stable interaction between uL14 and Rkm1 (Fig. 6 A, lanes 9–12), demonstrating that the methylation on the uL14 depends on Rkm1 but not Bcp1. Notably, the presence of Bcp1 did not dissociate uL14 from rkm1(Y273F) (Fig. 6 A, lanes 11–12). To test if Bcp1 could interact with methylated uL14, the supernatant fractions derived from lanes 7 and 8 of Fig. 6 A were collected and incubated with NTA beads to purify Bcp1. The interaction between uL14 and its methylation state was probed with antibodies, revealing that Bcp1 could interact with methylated uL14 (Fig. 6 B, lane 2). These findings suggest that Bcp1 is not involved in the

activation of Rkm1 but plays a pivotal role in disassembling methylated uL14.

We have proposed two possibilities for how the interaction of Bcp1 promotes the release of uL14. In the first scenario, uL14 is methylated at only one site when bound to Rkm1, and Bcp1 interaction triggers the activation of Rkm1, completing the methylation process. The second possibility is that the release of methylated uL14 from Rkm1 requires the involvement of Bcp1. To distinguish between these scenarios, we conducted a detailed examination of the methylation levels of uL14 using mass spectrometry. After the reaction with Rkm1 and SAM (Fig. 6 A,

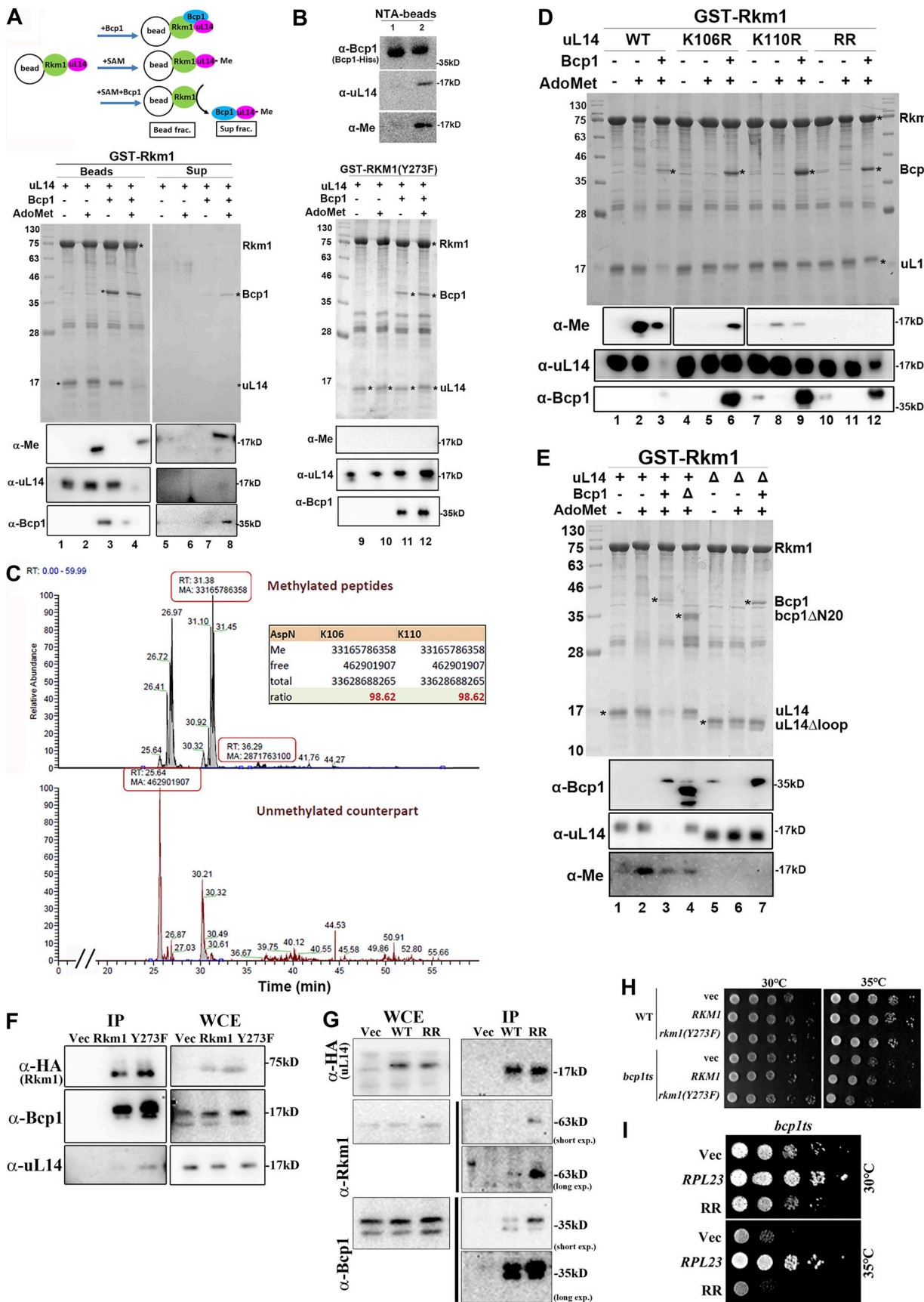

Figure 6. **Bcp1 triggers the release of Rkm1 from methylated uL14. (A)** GST-Rkm1/uL14 or GST-Rkm1(Y273F)/uL14 complex was immobilized on the glutathione beads. Buffer alone, containing 0.5 mM SAM or purified Bcp1, was added and incubated at 30°C for 80 min. After gently spinning, the supernatants

and beads were collected separately. TCA addition precipitated the proteins from supernatants (Sup). The glutathione beads were washed three times, and proteins were eluted in 1× SDS sample buffer. Proteins were analyzed by Coomassie blue staining and Western blotting with α-methylation (α-Me), α-uL14, and α-Bcp1 antibodies. **(B)** Bcp1-His$_6$ in the supernatants from reactions 7 and 8 in Fig. 6 A was applied for Ni-NTA purifications. The bound proteins on the NTA beads are shown as lanes 1 and 2, respectively. **(C)** uL14 was reacted with Rkm1 and SAM in vitro (Fig. 6 A, lane 2), and the in-gel digestion was performed with AspN. The intensities of methylated and unmethylated peptides were calculated, and the ratios are shown. **(D and E)** A complex of GST-Rkm1 with uL14, uL14(K106R), uL14(K110R), or uL14(RR) was immobilized on the glutathione beads. SAM (AdoMet) or with the purified Bcp1 were added (D). A complex of GST-Rkm1 with uL14 (+) or uL14Δloop (Δ) was immobilized on the glutathione beads. The purified Bcp1 (+) or bcp1ΔN20 (Δ) with SAM were added (E). The in vitro methylation assays proceeded as described above. Proteins that remained on the beads were shown in the Coomassie blue staining gel or detected by western blotting. **(F)** Rkm1-HA or rkm1(Y273F)-HA was immunoprecipitated and the associated proteins were detected by western blotting. **(G)** uL14-HA or uL14(RR)-HA was immunoprecipitated and the associated proteins were detected by western blotting. **(H)** WT or bcp1ts strains containing RKM1, or rkm1(Y273F) on 2 μ plasmids were normalized and serially diluted. Equal amounts of cells were spotted on the plates and incubated at the temperature indicated on the blot. **(I)** The growth tests of bcp1ts strain containing RPL23 or rpl23Δ(RR) plasmids. Source data are available for this figure: SourceData F6.

lane 2), uL14 was excised from the gel and subjected to digestion with four enzymes: AspN, GluC, LysC, and trypsin. The intensity ratio between methylated peptides and their unmethylated counterparts was analyzed using mass spectrometry data. In the case of AspN digestion, the methylation stoichiometry of K106 and K110 was ~98.6% (Fig. 6 C). Similar ratios were observed in other digestions: the methylation levels in GluC digestion were as higher as those detected in AspN (Fig. S9 A); the methylation levels in LysC digestion were about 80% (Fig. S9 B); and the methylation levels in trypsin digestion were about 60–70% (Fig. S9 C). Considering that nearly 86% of peptides were methylated at both K106 and K110 sites (Fig. S9 D), it is unlikely that Bcp1 functions as an activation factor but rather as a potential release factor for Rkm1.

### Bcp1, methylated uL14 at both sites, and correct assembly of the ternary complex are essential to trigger the disassembly of uL14 from Rkm1

To elucidate how Bcp1 influences methylation and release, one or both lysine sites of uL14 were mutated to arginine. While none of the mutants impacted the interaction with Rkm1 (Fig. 6 D), the methylation signals became undetectable when both lysine residues were mutated (Fig. 6 D, lanes 11 and 12). In the uL14(K106R) mutant, the methylation signal of K110 was absent in the presence of SAM (Fig. 6 D, lane 5) and exhibited a slight increase upon the addition of Bcp1 (Fig. 6 D, lane 6). In uL14(K110R) mutant, the methylation signal for K106 was weak and remained constant when Bcp1 was introduced (Fig. 6 D, compared lanes 8 and 9). Remarkably, both mutants persisted on Rkm1 even in the presence of Bcp1, and the methylation levels in either mutant were significantly lower than in the wild type. This suggests a potential synergistic relationship between methylation at the two sites.

Bcp1(ΔN20) and uL14Δloop were included in the methylation assay. Bcp1(ΔN20) still could form a ternary complex with Rkm1 and uL14. However, the methylation of uL14 significantly decreased with Bcp1(ΔN20) (Fig. 6 E, compare lane 2 and lane 4). On the other hand, the uL14Δloop, which contains methylation sites but lacks contact with Bcp1, maintained its interaction with Rkm1 (Fig. 6 E, lane 6) and formation of a ternary complex (Fig. 6 E, lane 7). However, even in the presence of Bcp1 and SAM, the uL14Δloop could not be methylated (Fig. 6 E, lanes 6 and 7).

To further demonstrate that uL14 methylation is critical for its release from Rkm1 in vivo, we also conducted immunoprecipitation experiments. Rkm1 could associate with Bcp1 and uL14 and the association was intensified in rkm1(Y273F) mutant (Fig. 6 F). uL14 and uL14(RR) were also immunopurified. In comparison with the wild type (WT), uL14(RR) exhibited higher signals for Bcp1 and Rkm1 interactions (Fig. 6 G). These data demonstrate that uL14 cannot be properly released from Rkm1 under methylation-defective situations. Additionally, overexpression of rkm1(Y273F) or uL14(RR) in bcp1ts impaired the growth (Fig. 6, H and I). In conclusion, faulty Bcp1 and uL14 impaired the proper Rkm1 methylation reaction, retaining uL14 on Rkm1. This step may function as a critical checkpoint for ensuring the quality of uL14.

## Discussion

### Bcp1 and Rkm1 are required for uL14 protection

This study shows that Bcp1, the chaperone for uL14, triggers the release of Rkm1 and Kap from uL14 and ensures stability. The proposed model (Fig 7) outlines the protection of the vital ribosomal protein uL14 through a series of events. Initially, uL14 is protected within a complex with Kap and Rkm1 during cotransport and undergoes methylation upon interaction with Rkm1. The subsequent interaction with Bcp1 triggers the release of Kap and Rkm1, with Bcp1 assuming the role of safeguarding uL14.

The unstructured loop (amino acids 40–55) of uL14 accommodates 60S subunits, exposing the C terminus as a binding site for Tif6 (Klinge et al., 2011; Wu et al., 2016). The extensions of ribosomal proteins are necessary for interacting with rRNAs or other proteins, highlighting their significance in translation and ribosome assembly (Kressler et al., 2017; Melnikov et al., 2012). Unstructured loops, like the ones in uL14, are highly susceptible to protease activity and require additional chaperones for protection to ensure efficient and accurate assembly of ribosomal proteins (Stelter et al., 2015). For example, Acl4, the dedicated chaperone for uL4, interacts with an extended internal loop at the C terminus of uL4, which is crucial for uL4 insertion into 60S subunits (Pillet et al., 2015; Stelter et al., 2015). Our structural data indicate that the loop (aa 40–55) of uL14 is shielded between Bcp1 and Rkm1 for protection. When subjected to partial proteolysis, individual uL14 or its complexes exhibited varying degrees of sensitivity. Notably, uL14 alone was the most susceptible to protease activity, while the addition of Rkm1 marginally increased its stability, and Bcp1 significantly enhanced its stability. Interestingly, uL14 demonstrated the highest stability when forming a ternary complex with Bcp1 and Rkm1. These findings suggest that Bcp1 and Rkm1

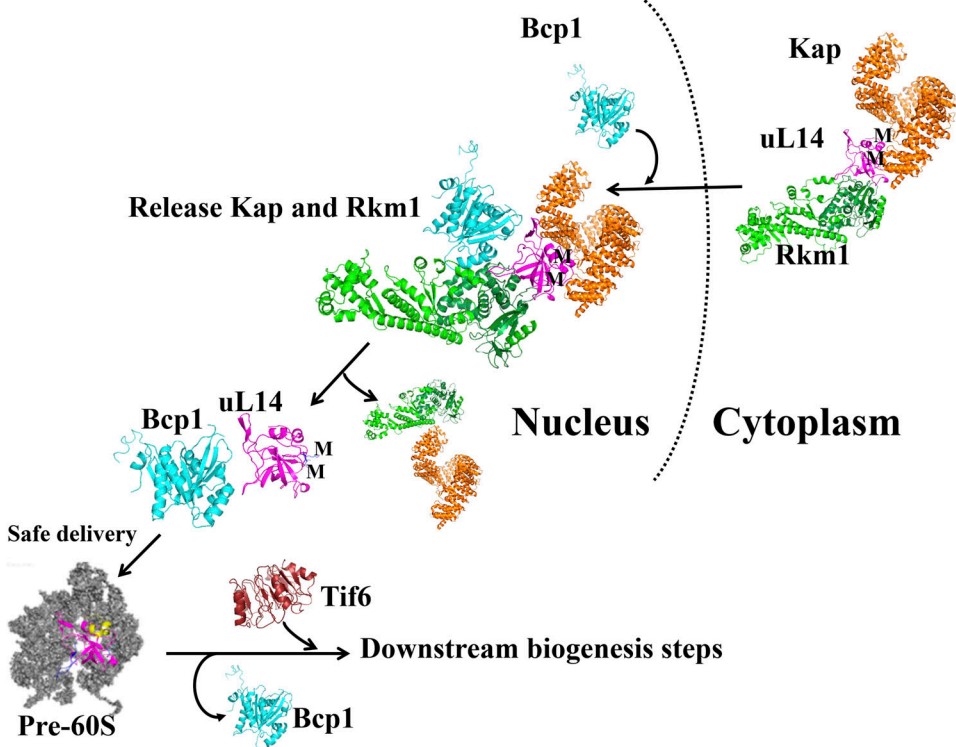

Figure 7. **Model figure Rkm1 interacts with uL14 in the cytoplasm and depends on uL14 for import.** Then Bcp1 releases the Rkm1 and karyopherins from uL14. Bcp1 interacts with uL14 possessing methylation modifications to ensure safe delivery to pre-60S.

synergistically protect uL14 and interact with it at different interfaces, corroborating our structural observations.

Our structural data demonstrate that Bcp1 and Rkm1 bind to uL14 at an interface similar to where uL14 binds to 60S (Fig. 4 A). Thus, these two factors are required to be released before uL14 incorporation.

Bcp1 is an essential gene, and although Rkm1 is also involved in uL14 protection, the deletion of Rkm1 showed similar growth rates as the WT under normal and stress conditions (Fig. S1 A). Interestingly, the *rkm1Δ* mutant exhibited an even higher growth rate in the presence of ribosome-targeting drugs, such as anisomycin and cycloheximide (Al-Hadid et al., 2016b). Deletion of *RKM1* could result in decreased growth when combined with the *bcp1ts* mutant (Fig. 1 A). The nascent uL14 level was decreased only in the *bcp1ts* but not in *rkm1Δ*, while the double mutant showed the lowest uL14 level. The discrepancy regarding why Rkm1 has a protection role without exhibiting any growth defects could be explained by the fact that Rkm1 and Kap interact with uL14 together in the cytoplasm, where Kap can also play a protective role for ribosomal proteins (Jäkel et al., 2002). This suggests a redundant function in protecting uL14 at this stage. However, it's important to note that Bcp1 is the sole nuclear chaperone of uL14. Furthermore, in the absence of Bcp1, uL14 cannot be released from the Kap/Rkm1 complex, preventing its loading onto the pre-60S subunit.

### The interaction with uL14 but not with Kap is crucial for Bcp1 to displace uL14 from Kap

The escortin Tsr2 interacts with the ESS in the ribosomal protein eS26, prompting a non-canonical RanGTP-independent disassembly of Tsr2. The deletion of the ESS of eS26 maintains its interaction with Kap but prevents its release by Tsr2 (Schütz et al., 2018). When comparing the amino acid sequences of uL14 between *E. coli* and yeast, a 47.2% similarity is observed (Fig. S10 A) (Madeira et al., 2022). In structural comparison, the N-terminus and internal loop of yeast uL14 (PDB accession no. 5H4P) (Ma et al., 2017) are longer than those of *E. coli* uL14 (PDB accession no. 1ML5) (Klaholz et al., 2003) (Fig. S10 B). In the ternary structure, the N terminus is not in contact with Bcp1 or Rkm1 (Fig. 4 A), but Bcp1 and Rkm1 sandwich the loop region. Consistent with this, uL14(Δloop) loses its association with Bcp1 but not with Kap, and it cannot be released from Kap by Bcp1. Our data also indicate that the N-terminus of Bcp1 is critical for Kap interaction but not essential for displacing uL14 from Kap (Fig. 5). This information emphasizes the significance of the interaction between escortin and the ribosomal protein in separating uL14 from Kap.

### Methylation on uL14 serves as protection and quality surveillance

Posttranslational modifications are important to regulate protein–protein interactions and functions. Methylation, a prevalent modification, primarily occurs on lysine and arginine residues but can also be observed on histidine and carboxyl groups of proteins (Grillo and Colombatto, 2005). While the physiological functions of methylation have been extensively studied in histones, it is worth noting that non-histone proteins, including numerous ribosomal proteins, are also targets of methylation (Pang et al., 2010). The functions of modifications on the ribosomal proteins are not clear yet, but they have been shown to adjust interactions

with rRNAs, modulate stress responses, influence associations with other proteins, and fine-tune translation (Huang and Berger, 2008). In *Schizosaccharomyces pombe*, methyltransferase Set13, Set11, and Rmt3, responsible for modifying eL42 (Rpl42), uL11 (Rpl12), and uS5 (Rps2), respectively, play essential roles in ribosomal subunit production (Perreault et al., 2009; Sadaie et al., 2008; Shirai et al., 2010). In humans, the failure of uS3 (S3) and eS10 (S10) to undergo methylation results in their inability to localize to the nucleolus and incorporate properly into the ribosome (Shin et al., 2009). Histidine methylation of uL3 (Rpl3) is a requirement for 60S subunit assembly in yeast (Webb et al., 2010). An investigation into 10 ribosomal protein methyltransferases in *Saccharomyces cerevisiae* revealed that the deletion mutants, *rkm1Δ*, *ntm1Δ*, *rmt1Δ*, and *rmt2Δ*, exhibited a deficiency in 60S subunits. These enzymes are important for translation elongation fidelity or termination efficiency (Al-Hadid et al., 2016b).

In this study, we have identified the critical role of methylation in safeguarding and monitoring protein quality. To further examine the methylation at lysine residues K106 and K110 on uL14, one or two lysine sites were mutated to Arg, and the double mutant displayed a loss of methylation signals. Specifically, the uL14(K106R) mutant resulted in the complete abolition of the methylation signal, while the uL14(K110R) mutant exhibited a significant reduction in methylation (Fig. 6 D). This suggests that methylation at K106 is a prerequisite for downstream methylation at K110, and there might be a synergistic relationship between the methylations at these two sites. Analogous to phosphorylation in kinase, where phosphorylation at a priming site is required for downstream phosphorylation events. The hierarchical phosphorylation is critical for tight control and coordination between different pathways (Jeschke et al., 2018). Notably, the addition of Bcp1 could only marginally increase methylation at K110 in the uL14(K106R) mutant, and it had no impact on the methylation level at K106 in the uL14(K110R) mutant. This suggests that the binding of Bcp1 might reposition K110 toward the enzymatic activity center of Rkm1.

Another intriguing observation is that Bcp1 is indispensable for dissociating Rkm1 from uL14, but this requirement does not extend to mutant uL14. WT uL14 remains bound to Rkm1 even when the lysine residuals have been methylated and only dissociates when Bcp1 is present. Both uL14(K106R) and uL14(K110R) mutants persistently associate with Rkm1 in vitro, even in the presence of Bcp1 and SAM. uL14Δloop, lacking contact with Bcp1, still forms a complex with Rkm1 and Bcp1, suggesting that the contact might differ in the dimeric and trimeric forms. Although uL14Δloop contains methylation sites, it couldn't be methylated and retained in the ternary complex stage. The correct assembly of the Bcp1, uL14, and Rkm1 complex is essential to initiate the transfer of uL14 to Bcp1, thereby ensuring the stability of uL14. Additionally, this mechanism serves as a quality checkpoint, preventing the loading of mutant uL14 onto 60S subunits.

# Materials and methods
## Strains, plasmids, and media
All *S. cerevisiae* strains used in this study are listed in Table S1. Unless otherwise indicated, all strains were grown at 30°C in a rich medium (yeast extract peptone) or synthetic dropout medium containing 2% glucose. The plasmids used in this study are listed in Table S2.

## SDS-PAGE and Western blotting
The proteins were separated using 8%, 10%, or 15% SDS-PAGE gels based on their molecular weight, utilizing a protein electrophoresis tank (Hoefer). Subsequently, the proteins were transferred onto PVDF membranes (Bio-Rad) using a Trans-Blot SD semi-dry transfer device (Bio-Rad). The membranes were blocked with 10% non-fat milk for at least 30 min. To identify the target proteins during Western blotting, the membranes were incubated overnight at 4°C with primary antibodies, which were diluted in TBST buffer (20 mM Tris, 150 mM NaCl, 0.1% Tween 20) at a concentration ranging from 1:2,000 to 1:5,000. The anti-myc antibody was purified from MYC 1-9E10.2 [9E10] (ATCC CRL1729) in this lab. The anti-methylation (ab23366; Abcam), anti-TAP (CAB1001; Thermo Fisher Scientific), anti-HA (ARG62338; Arigo), anti-His$_6$ (HIT001M; Bioman), and anti-GFP (11814460001; Sigma-Aldrich) antibodies were purchased from the companies. Anti-actin antibody was generated in Dr. Fang-Jen Lee's laboratory (Tsai et al., 2008) and provided by the Taiwan Yeast Resource Center at the College of Medicine, National Taiwan University, Taipei, Taiwan. Anti-Bcp1, anti-uL14, anti-eL8, anti-eS24, anti-eL43, and anti-Rkm1 antibodies were generated in this lab (Ting et al., 2017; Yang et al., 2016). After three washes, the membranes were then incubated with a horseradish peroxidase-conjugated secondary antibody (#7074, anti-rabbit IgG; #7076, anti-mouse IgG; Cell Signaling Technology) for 60 min at room temperature. Protein signals were visualized using Clarity ECL Substrate (Bio-Rad) and images were captured with MultiGel-21 (TopBio).

## Analysis of nascent uL14 proteins by ultracentrifugation
Cultures were grown to an OD$_{600}$ of 0.4–0.5 in the medium. Protein extracts were prepared by vortexing with glass beads in extraction buffer (50 mM NaCl, 20 mM Tris, pH 7.5, 6 mM MgCl$_2$, 10% glycerol, 0.1% NP-40, 1 mM PMSF, 1 μM leupeptin, and 1 μM pepstatin A). 800 μl of protein extracts was centrifuged at 80,000 rpm in a rotor (MLA130; Beckman Coulter) at 4°C for 60 min. Free proteins and ribosomes were separated into supernatants and pellets, respectively. Proteins from the supernatants were precipitated with 10% trichloroacetic acid (TCA) and detected by Western blotting.

## Sucrose gradient analysis
For polysome profile assays, cultures were collected at an OD$_{600}$ of 0.2–0.3. 50 μg/ml of cycloheximide was added before cell collection. Polysome lysis buffer (10 mM Tris-HCl, pH 7.5, 100 mM KCl, 10 mM MgCl$_2$, 6 mM β-mercaptoethanol, and 200 μg/ml cycloheximide) was used for the preparation of protein extracts. 10.5 OD$_{260}$ units of protein extracts were loaded onto linear 7–47% sucrose gradients and spun at 40,000 rpm in a rotor (SW40; Beckman) for 2.5 h. Gradient fractions were collected on a density gradient fraction system (Brandel), continuously measuring absorbance at 254 nm. 10% TCA was added to each fraction to precipitate proteins. Dissolved

the protein pellets were in 1× SDS sample buffer. Samples were resolved by SDS-PAGE and detected by Western blotting.

## Microscopy

Overnight cultures were diluted in fresh media to an $OD_{600}$ of 0.1 and were incubated for another 2 h at 30°C. For *ts* (temperature sensitive) mutant strains, cells were shifted to 37°C for 2 h before assay. Fluorescence was visualized on a microscope (AxioScope A1; Zeiss) fitted with a Plan Apochromat 100× 1.40 NA DIC objective and a digital microscopy camera (AxioCam MRm Rev. 3) controlled with AxioVision LE module Fluorescence Lite (Zeiss). Images were prepared using Photoshop (version 7.0; Adobe).

## Immunoprecipitation

For immunoprecipitations, cultures were grown to an $OD_{600}$ of ∼0.5 in a selective medium. Before cell harvest, the *bcp1ts* (temperature-sensitive) mutant was shifted to 37°C for 2 h, or 2% glucose was added in *GAL::RPL23* for 4 h. Cells were resuspended in IP buffer (20 mM Tris pH 7.5, 50 mM NaCl, 6 mM $MgCl_2$, 10% glycerol, 1 mM PMSF, and 1 mM leupeptin), lysed by vortexing with glass beads for 30 s with a 1-min interval on ice for six times. α-c-myc antibody was added to normalized protein extracts and incubated for 2 h at 4°C. Protein A agarose beads (L00210; GenScript) or IgG beads (GE17-0969-01; Merck) were subsequently added and incubated for another hour. After three washes, proteins were eluted in 1× Laemmli sample buffer and detected by Western blotting.

## Expression and purification of recombinant Bcp1-His₆, uL14, and Rkm1-His₆ protein

*S. cerevisiae* Bcp1, uL14, and Rkm1 were expressed in the *E. coli* BL21(DE3) strain using 0.5 mM IPTG at 16°C overnight and purified using Ni affinity chromatography. The Bcp1-His₆ and uL14 bacterial pellets were combined with PBS lysis buffer and lysed by the French press. Rkm1 bacterial pellet was resuspended in PBS lysis buffer and lysed by the French press. The cell lysates were centrifuged at high speed, collected in the supernatant, and loaded onto the open column containing the Ni-NTA resin separately. Both columns were washed with 5 C.V. wash buffer (PBS buffer). The Bcp1/uL14 protein complex was eluted with elution buffer (100 mM NaCl, 50 mM Na citrate, pH 5.5, 5% glycerol). The Rkm1 protein was eluted with elution buffer (100 mM NaCl, 50 mM Tris, pH 7.5, 300 mM Imidazole, 5% glycerol). Further purification will be conducted by Superdex 200 size exclusive column (GE). For Bcp1/uL14/Rkm1 complex preparation, those proteins were mixed and purified using HiLoad 16/600 Superdex 200 column (GE). The protein complex with the peak corresponding to the appropriate molecular weight was collected.

## Chemical crosslinking coupled to mass spectrometry (XL-MS) of Bcp1/uL14/Rkm1 complex

The crosslinker glutaraldehyde (Sigma-Aldrich) was used to crosslink intermolecular lysine residues. Purified Bcp1/uL14/Rkm1 complex at 1 mg/ml at PBS buffer was incubated with 0.05% glutaraldehyde for 20 min at room temperature. The reaction was quenched by 0.1 M Tris-HCl. The crosslinked complexes and uncrosslinked proteins were separated by SDS-PAGE, followed by in-gel digestion using Lys-C protease and chymotrypsin (Thermo Fisher Scientific). The digested peptide mixture was desalted, lyophilized and then stored at –20°C prior to LC-MS/MS analysis.

The LC-MS/MS analysis was performed on an Orbitrap Fusion mass spectrometer (Thermo Fisher Scientific) equipped with EASY-nLC 1200 system (Thermo Fisher Scientific) and EASY-Spray HPLC column (75 μm I.D. × 150 mm, 3 μm, 100 Å) and ion source (Thermo Fisher Scientific). The chromatographic separation was performed using 0.1% formic acid in water as mobile phase A and 0.1% formic acid in 80% acetonitrile as mobile phase B operated at the flow rate of 300 nl/min. The LC gradient was employed from 2% buffer B at 2 min to 40% buffer B at 40 min. Electrospray voltage was maintained at 1.8 kV and the capillary temperature was set at 275°C. Full MS survey scans were executed in the mass range of m/z 320–1,600 (AGC target at $5 \times 10^5$) with lock mass, resolution of 120,000 (at m/z 200), and maximum injection time of 50 ms. The MS/MS was run in top speed mode with 3 s cycles; while the dynamic exclusion duration was set to 60 s with a 10 ppm tolerance around the selected precursor and its isotopes. The precursor ion isolation was performed with mass selecting iontrap and the isolation window was set to m/z 3.0. Monoisotopic precursor ion selection was enabled and 1+ charge state ions were rejected for MS/MS. The MS/MS analyses were carried out with the collision-induced dissociation (CID) mode with a collision energy of 35%. The maximum injection time for spectra acquisition was 100 ms and the automatic gain control (AGC) target values for MS/MS scans were set at $5 \times 10^4$.

Acquired MS raw data were converted as mgf format by msConvert (version 3.0.18165; ProteoWizard), then analyzed using MassMatrix (ver. 3.10) for MS/MS ion search of crosslinked peptides. The search configuration included the precursor ion tolerance of 10 ppm, product ion tolerance of 0.5 Da, the maximum number of PTM/peptide was 2, minimum peptide length was 5, minimum PP score was 2.5, minimum PPtag score was 1.2, the maximum number of matches/spectrum was 2, the maximum number of combinations/match was 2, and the maximum number of crosslinks/peptide was 2.

## Computational simulation for ternary complex model

All crosslink-guided docking calculations were performed using the ROSETTA (Kahraman et al., 2013). The initial structure of Rkm1 was predicted with AlphaFold (Jumper et al., 2021). The structure of Bcp1 (PDB accession no. 7C4H) (Lin et al., 2020) was obtained from the protein data bank, whereas the structure of uL14 can be extracted from published pre-60S complex (PDB accession no. 5H4P) (Ma et al., 2017). First, we applied the crosslink data as distance restraints force for global docking producing 1,000 models. 15 Å is the cutoff for the distance between all the identified lysine pairs in each of the models. All the global docking models were filtered by Xwalk (Kahraman et al., 2011) to determine which models satisfy the most crosslinks. Then we selected 50 models with the lowest energy scores and only models with a sufficiently large binding interface by using the NACCESS (Hubbard and Thornton, 1993).

Second, we performed a Quality-Threshold (QT) clustering and local refinement docking calculation on each of the three cluster representatives ~3 × 50 models, filtered, and computed binding interface size again. Finally, we calculated the RMSD and contact frequency to select the lowest-scoring model from the clusters as the best prediction from the entire docking run.

## Model relaxation by molecular dynamics Simulation

The crosslink-based docking model was taken as a starting coordinate for MD simulations. MD Simulations were performed based on a force field Amber ff14SB (Maier et al., 2015), and the residue charges were calculated based on the libraries in the Amber 16 package (Case et al., 2016). Periodic boundary conditions were imposed with box lengths of 128.10 × 114.42 × 140.48 Å³, and containing 946 amino acids and 54,214 TIP4P water models. The SHAKE algorithm was implemented to constrain the covalent bond, including hydrogen atoms. The MD System underwent a 25 ns annealing process under the constant pressure of 1.0 bar with equilibrated steps from 0 to 300 K. A Langevin thermostat was used to maintain the system temperature by controlling the collision frequency at 1 ps$^{-1}$ to the target temperature of 300 K. After the annealing step, 20-ns MD simulations were carried out in the canonical ensemble (NVT) with the Langevin thermostat to maintain the system temperature.

## Conformational morphing of complex model

The crosslink-based docking model was taken as a starting conformation, and the MD relaxation model was taken as an end conformation. The RigiMol method was used to create the trajectories from the starting conformation to the end conformation. The refinement step was set as three cycles and generated 30 output states. All the simulation was done using Pymol v2.4.1 (The PyMOL Molecular Graphics System, Version 2.4.1 Schrödinger, LLC.).

## CryoEM sample preparation and data collection

The purified crosslinked Bcp1/uL14/Rkm1 complex was stored in a PBS buffer. The protein complex samples were applied on a glow-discharged Quantifoil holey carbon grid (1.2/1.3, 200 mesh) coated with graphene-oxide. The grids were blotted for 4 s at 100% humidity with 4°C and plunge-frozen in liquid ethane cooled by liquid nitrogen using a Vitrobot Mark IV system (Thermo Fisher Scientific). Cryo-EM data for all samples were acquired on a Titan Krios electron microscope (Thermo Fisher Scientific) at 300 KeV, equipped with a Quantum K3 Summit direct electron detector (Gatan) at Academia Sinica cryo-EM facility, with energy selecting slit of 18 eV. Automatic data acquisition was carried out using EPU software (Thermo Fisher Scientific) at a nominal magnification of 105,000× corresponding to a calibrated 0.83-pixel size. Movies of 50 frames, corresponding to a total dose of 50 e⁻Å⁻², were collected in super-resolution mode at a dose rate of 1 e⁻Å⁻² per frame, and the internal defocus range for the sample was between −1 and −2 μm.

## Cryo-EM structure reconstruction

The image processing flowchart is summarized in Fig. S2. 5,484 dose-fractionated movies were dominated to motion correction using the program MotionCor2 (Zheng et al., 2017) with dose weighting and then using the program Patch CTF estimation (multi) (Punjani et al., 2017) for estimate defocus values for all movie frames. 1,931,871 particles were extracted for 2D classifications and 1,189,719 particles were selected for ab initio reconstruction in cryoSPARC (Punjani et al., 2017). Due to the nature of structural heterogeneity, only a low-resolution envelope was generated and the resolution was estimated using d99, which estimates the resolution related to map details in the real space (Afonine et al., 2018). The envelope generated by cryo-EM is in agreement with the ternary complex generated by cross-linking and simulation.

## Methylation assays on uL14

The GST-Rkm1/uL14 complex was purified with the glutathione beads for methylation assay. Buffer alone, purified Bcp1, or with 0.5 mM AdoMet (Sigma-Aldrich) was added and incubated at 30°C for 80 min. The methylation level on the uL14 was detected by an anti-methylation antibody (Abcam). To measure the methylations of uL14, the proteins after in vitro methylation reaction were resolved in SDS-PAGE. A gel slice containing uL14 was digested with four proteases, AspN, GluC, LysC, and trypsin to cover the entire sequence. The peptide mixtures were detected by LC-ESI-MS on an Orbitrap Fusion mass spectrometer (Thermo Fisher Scientific) equipped with EASY-nLC 1200 system (Thermo Fisher Scientific) and EASY-spray source (Thermo Fisher Scientific). The sum of the peak area of the extracted ion chromatogram of each identified peptide obtained the total intensity of peptides. The total intensity of methylated peptides and its unmethylated counter peptides was calculated for the methylation ratio.

## Statistical analysis

The biological replicates are indicated in figure legends. Data were plotted and analyzed in Excel. A two-tailed Student's $t$ test was performed against the control to assess statistical significance ($*P < 0.05$; $**P < 0.01$).

## Online supplemental material

Fig. S1 shows that the Bcp1 mutant did not change the protein level of Rkm1 and vice versa; Fig. S2 shows the flow-chart of Bcp1/Rpl23/Rkm1 cryoEM data process; Fig. S3 shows the XL-MS results of the ternary complex, Bcp1(K150)-Rkm1(K334); Fig. S4 shows the XL-MS results of the ternary complex, Bcp1(K187)-Rkm1(K302); Fig. S5 shows the XL-MS results of the ternary complex, uL14(K40)-Rkm1(K334); Fig. S6 shows the XL-MS results of the ternary complex, uL14(K63)-Rkm1(K78); Fig. S7 shows the XL-MS results of the ternary complex, Bcp1(K254)-uL14(K64); Fig. S8 shows the sequence alignment and comparison of the SET domain of Rkm1 and other methyltransferases; Fig. S9 shows the MS analysis of methylation sites of uL14 by Rkm1 in the in vitro methylation reaction; Fig. S10 shows comparisons of E. coli and yeast uL14. Table S1 shows the yeast strains used in this study; Table S2 lists the plasmids used in this study. In Video 1 molecular dynamics simulation shows that protein dynamics enable the movement of Lys106 and Lys110 residues to the Rkm1 active site for methylation.

## Data availability

All data generated or analyzed during this study are included in this published article and its supplementary information files.

The rest of the data generated in this study are available from the corresponding authors upon reasonable request.

## Acknowledgments

We thank the staff of the Human Disease Modeling Center at the First Core Labs, National Taiwan University College of Medicine, for sharing bioresources. We acknowledge the mass spectrometry technical research services from NTU Consortia of Key Technologies and NTU Instrumentation Center. We thank the staff at the centers below for technical support: Join Center for Instruments and Researches, College of Bioresources and Agriculture of National Taiwan University; the Technology Commons, College of Life Science, National Taiwan University. The cryo-EM data were collected at the Academia Sinica Cryo-EM Center (ASCEM, supported by Taiwan Protein Project Grant number AS-KPQ-105-TPP and AS-KPQ-109-TPP2 and AS Grant number AS-CFII-108-110) and data were processed at the Academia Sinica Grid-computing Center (ASGC, supported by Academia Sinica). We are grateful to the GRC Mass Core Facility of Genomics Research Center, Academia Sinica for MS data collection.

This work was financially supported by the National Science and Technology Council of Taiwan (MOST 109-2313-B-002-023-MY3 and NSTC 112-2313-B-002-049-MY3 to K.-Y. Lo) and (MOST 111-2311-B-001-009 to M.-C. Ho). Open Access funding provided by the National University of Taiwan.

Author contributions: M.-C. Yeh: Conceptualization, Data curation, Formal analysis, Methodology, Validation, Visualization, Writing—original draft, N.-H. Hsu: Investigation, Methodology, H.-Y. Chu: Data curation, Formal analysis, Investigation, Methodology, Validation, Visualization, C.-H. Yang: Formal analysis, Methodology, Writing—original draft, P.-H. Hsu: Data curation, Formal analysis, C.-C. Chou: Formal analysis, Investigation, Writing—review & editing, J.-T. Shie: Conceptualization, Investigation, Validation, Visualization, W.-M. Lee: Investigation, Validation, M.-C. Ho: Conceptualization, Data curation, Funding acquisition, Methodology, Project administration, Resources, Supervision, Validation, Writing—original draft, K.-Y. Lo: Conceptualization, Funding acquisition, Project administration, Supervision, Writing—original draft, Writing—review & editing.

Disclosures: The authors declare no competing interests exist.

Submitted: 28 June 2023

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

# Supplemental material

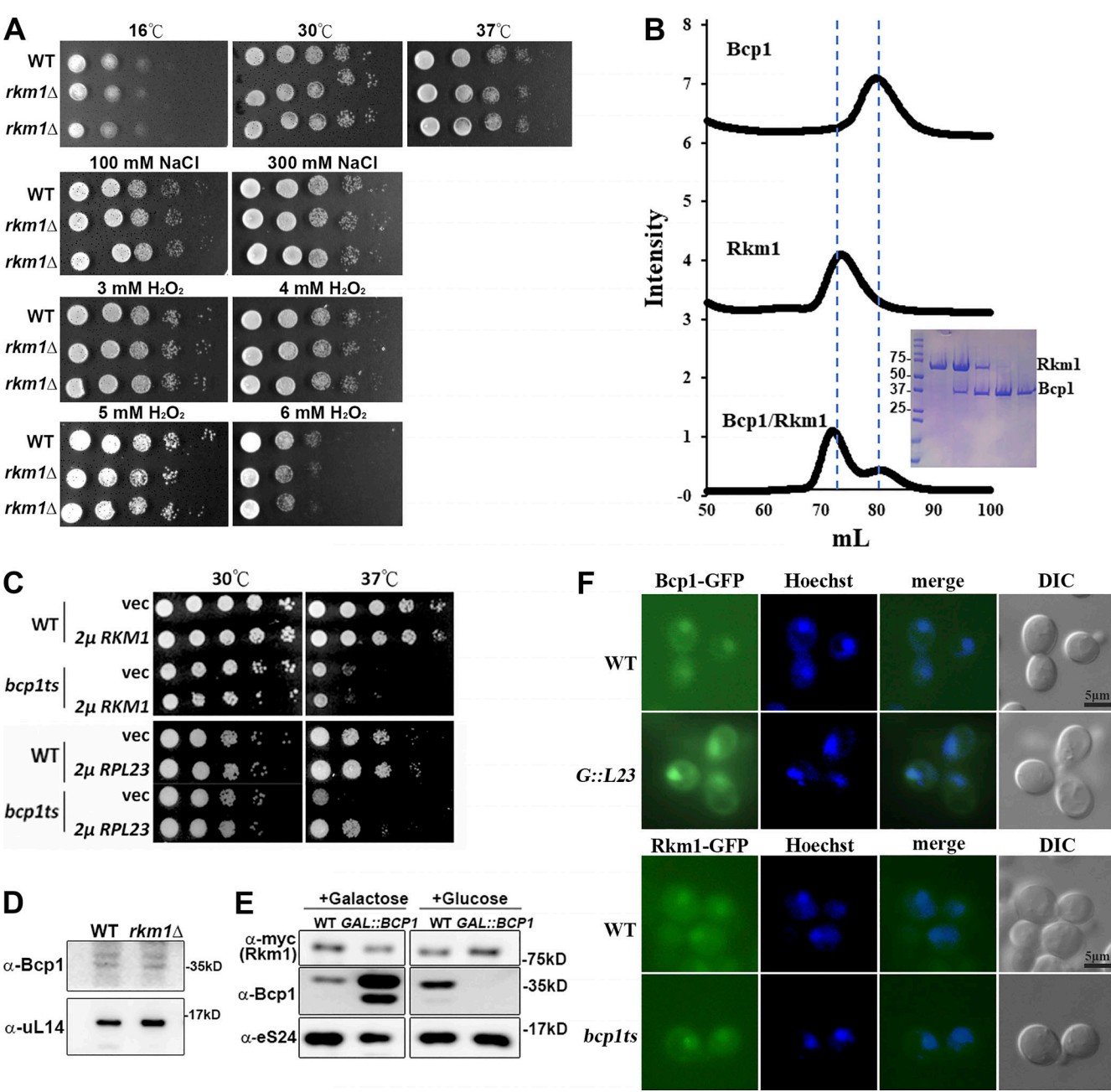

Figure S1. **Bcp1 mutant did not change the protein level of Rkm1 and vice versa. (A)** The growth tests of *rkm1Δ* at different conditions. **(B)** The size exclusion chromatography (left). The result of Coomassie blue gels (right). The y-axis is the normalized intensity of UV280 nm. The x-axis is the elution volume of the Superdex 200 column. **(C)** The growth tests of WT and *bcp1ts* containing vector, 2µ *RKM1*, or 2µ *RPL23*. **(D)** The protein level of Bcp1 was detected in WT and *rkm1Δ*. **(E)** The protein level of Rkm1 was detected under overexpression or depletion of Bcp1. **(F)** The localization of Bcp1-GFP was visualized in wild-type and *GAL::RPL23*. Overnight cells were subcultured in a medium containing 2% galactose for 2 h and 2% glucose was added for another 4 h before examination with fluorescence microscopy. To visualize the localization of Rkm1-GFP in wild-type and *bcp1ts* at log phase were shifted to 37°C for 2 h. Source data are available for this figure: SourceData FS1.

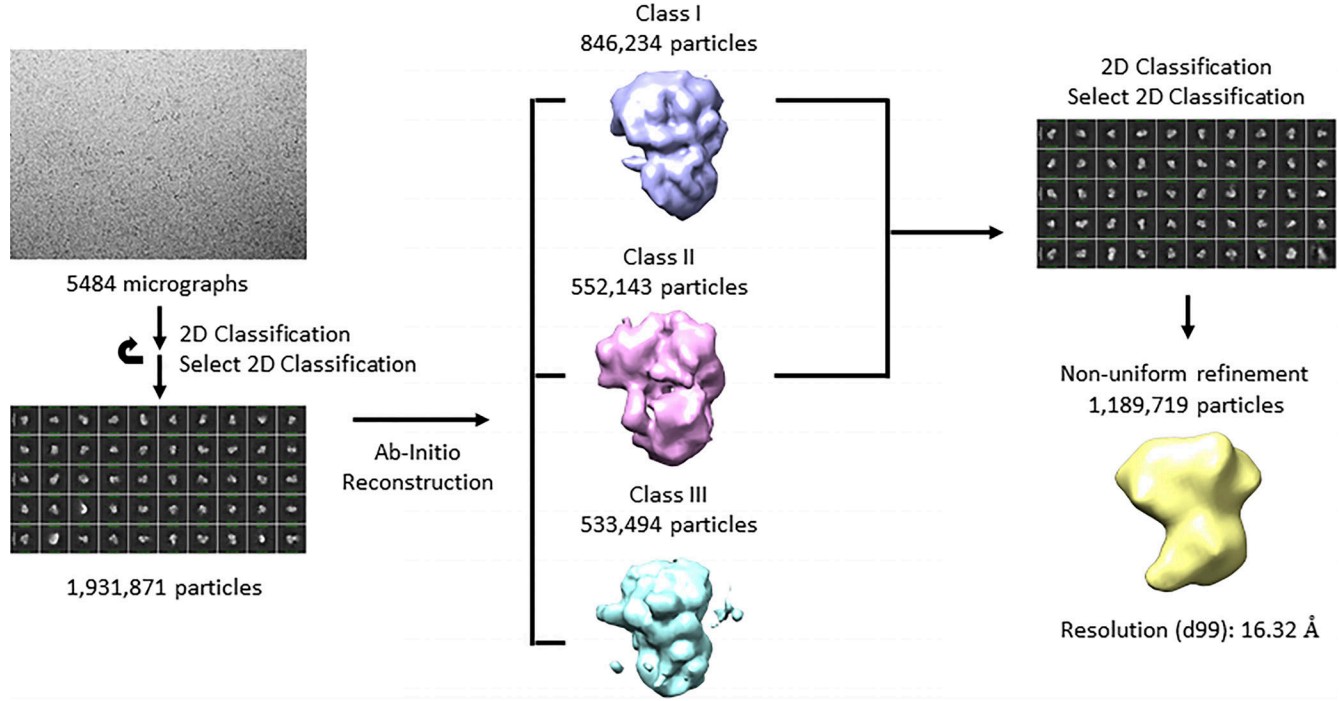

Figure S2.  **The flowchart of Bcp1/Rpl23/Rkm1 cryoEM data process.** The workflow for structure determination. A representative cryo-EM micrograph and a representative 2D class average are shown. Initial 3D models were separated into three classes. By applying only classes 1 and 2 for non-uniform refinement, the final model is at 16.32 Å.

## Bcp1 K150 - RKM1 K334

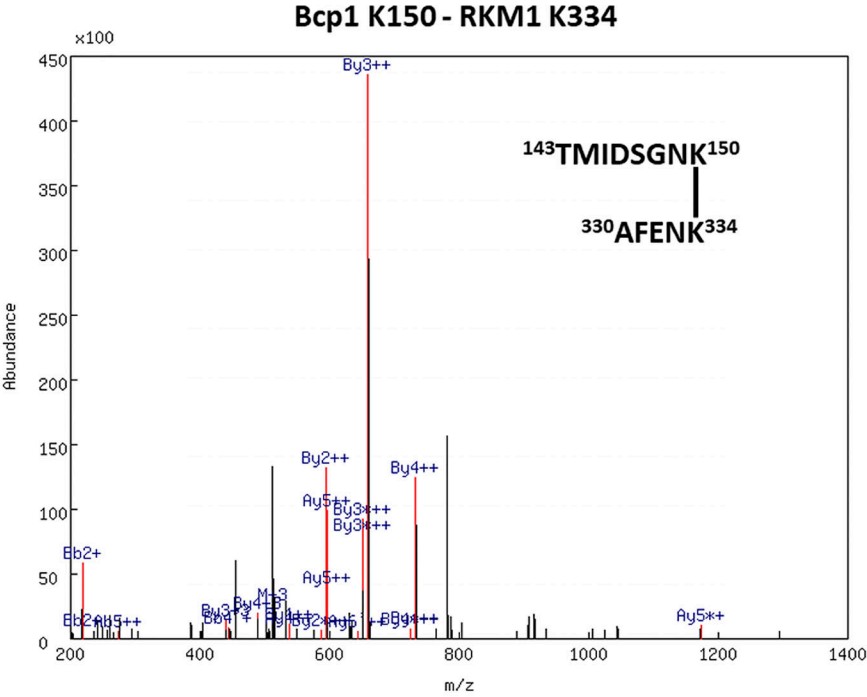

$^{143}$TMIDSGNK$^{150}$

$^{330}$AFENK$^{334}$

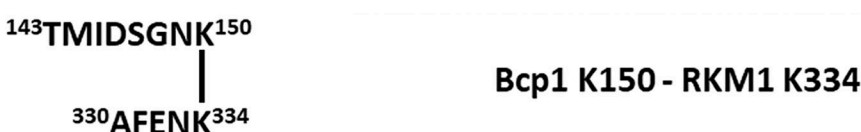

### Bcp1 K150 - RKM1 K334

### Bcp1, $^{143}$TMIDSGNK$^{150}$

| # | b'+3 | b*+3 | b+3 | b'++ | b*++ | b++ | b'+ | b*+ | b+ | seq | y'+3 | y*+3 | y+3 | y'++ | y*++ | y++ | y'+ | y*+ | y+ | # |
|---|------|------|-----|------|------|-----|-----|-----|-----|-----|------|------|------|------|------|------|------|------|------|---|
| 1 | 28.69 | -- | 34.69 | 42.53 | -- | 51.53 | 84.04 | | 102.05 | T | 506.91 | 507.24 | 512.92 | 759.87 | 760.36 | 768.87 | 1518.73 | 1519.71 | 1536.74 | M |
| 2 | 72.37 | -- | 78.37 | 108.05 | -- | 117.05 | 215.08 | | 233.10 | M | 473.23 | 473.56 | 479.23 | 709.34 | 709.83 | 718.35 | 1417.68 | 1418.66 | 1435.69 | 7 |
| 3 | 110.06 | -- | 116.06 | 164.59 | -- | 173.59 | 328.17 | | 346.18 | I | 429.55 | 429.88 | 435.55 | 643.82 | 644.31 | 652.83 | 1286.64 | 1287.62 | 1304.65 | 6 |
| 4 | 148.40 | -- | 154.41 | 222.10 | -- | 231.11 | 443.20 | | 461.21 | D | 391.86 | 392.18 | 397.86 | 587.28 | 587.77 | 596.29 | 1173.55 | 1174.54 | 1191.56 | 5 |
| 5 | 177.41 | -- | 183.42 | 265.62 | -- | 274.62 | 530.23 | | 548.24 | S | 353.51 | 353.84 | 359.52 | 529.77 | 530.26 | 538.77 | 1058.53 | 1059.51 | 1076.54 | 4 |
| 6 | 196.42 | -- | 202.42 | 294.13 | -- | 303.13 | 587.25 | | 605.26 | G | 324.50 | 324.83 | 330.51 | 486.25 | 486.74 | 495.26 | 971.49 | 972.48 | 989.51 | 3 |
| 7 | 234.44 | 234.76 | 240.44 | 351.15 | 351.64 | 360.16 | 701.29 | 702.28 | 719.30 | N | 305.50 | 305.82 | 311.50 | 457.74 | 458.23 | 466.75 | 914.47 | 915.46 | 932.48 | 2 |
| | | | | | | | | | | K | 267.48 | 267.81 | 273.49 | 400.72 | 401.21 | 409.72 | 800.43 | 801.41 | 818.44 | 1 |

### RKM1, $^{330}$AFENK$^{334}$

| # | b'+3 | b*+3 | b+3 | b'++ | b*++ | b++ | b'+ | b*+ | b+ | seq | y'+3 | y*+3 | y+3 | y'++ | y*++ | y++ | y'+ | y*+ | y+ | # |
|---|------|------|-----|------|------|-----|-----|-----|-----|-----|------|------|------|------|------|------|------|------|------|---|
| 1 | -- | -- | 24.69 | | -- | 36.53 | | -- | 72.04 | A | -- | | | | | | | | | |
| 2 | -- | -- | 73.71 | | -- | 110.06 | | -- | 219.11 | F | 483.23 | 483.56 | 489.24 | 724.35 | 724.84 | 733.35 | 1447.69 | 1448.67 | 1465.70 | 4 |
| 3 | 110.72 | -- | 116.72 | 165.58 | -- | 174.58 | 330.14 | | 348.16 | E | 434.21 | 434.54 | 440.22 | 650.81 | 651.31 | 659.82 | 1300.62 | 1301.60 | 1318.63 | 3 |
| 4 | 148.73 | 149.06 | 154.74 | 222.60 | 223.09 | 231.60 | 444.19 | 445.17 | 462.20 | N | 391.20 | 391.53 | 397.20 | 586.29 | 586.78 | 595.30 | 1171.58 | 1172.56 | 1189.59 | 2 |
| | | | | | | | | | | K | 353.18 | 353.51 | 359.19 | 529.27 | 529.76 | 538.28 | 1057.53 | 1058.52 | 1075.55 | 1 |

Figure S3. **The XL-MS results of the ternary complex, Bcp1(K150)-Rkm1(K334).**

## Bcp1 K187 - RKM1 K302

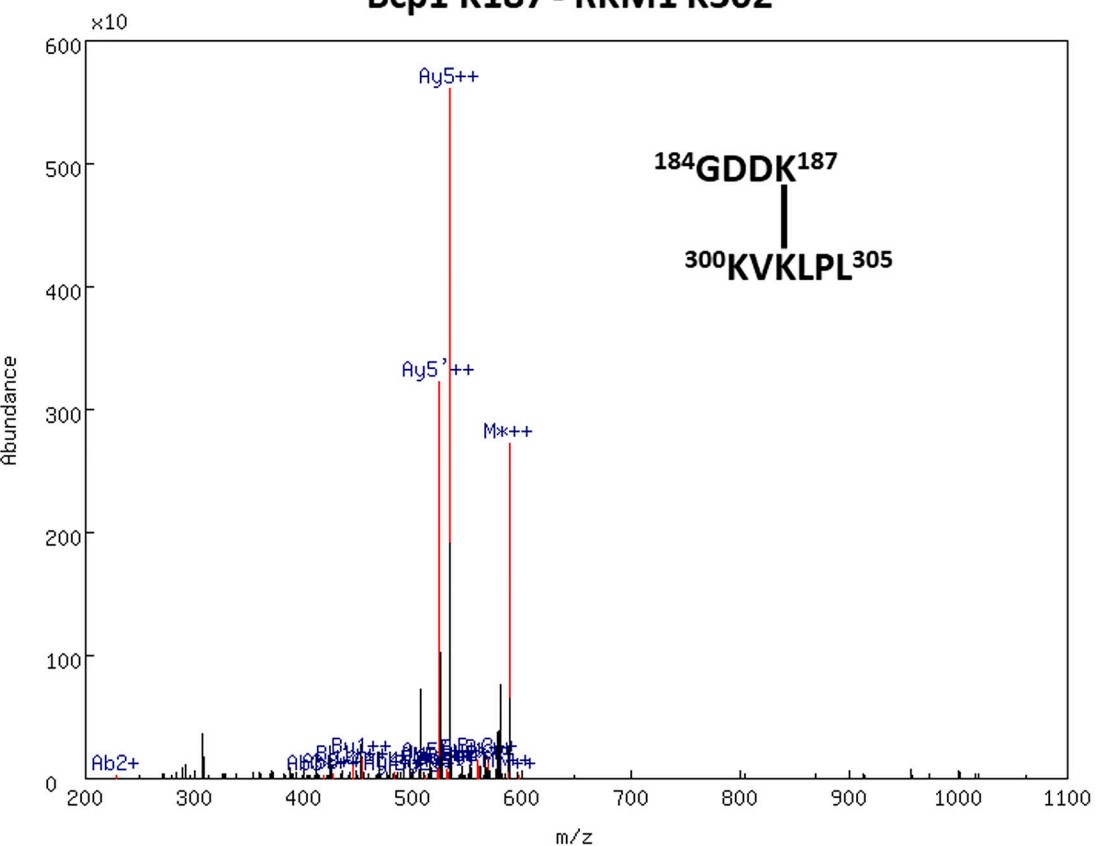

## Bcp1, $^{184}$GDDK$^{187}$

| # | b'++ | b*++ | b++ | b'+ | b*+ | b+ | seq | y'++ | y*++ | y++ | y'+ | y*+ | y+ | # |
|---|------|------|-----|-----|-----|----|-----|------|------|-----|-----|-----|----|---|
| 1 | — | — | 29.52 | | — | 58.03 | G | — | — | — | — | — | — | |
| 2 | 78.03 | — | 87.03 | 155.05 | — | 173.06 | D | 560.34 | 560.83 | 569.35 | 1119.68 | 1120.66 | 1137.69 | 3 |
| 3 | 135.54 | — | 144.54 | 270.07 | — | 288.08 | D | 502.83 | 503.32 | 511.83 | 1004.65 | 1005.63 | 1022.66 | 2 |
| | — | — | — | — | — | — | K | 445.32 | 445.81 | 454.32 | 889.62 | 890.61 | 907.63 | 1 |

## RKM1, $^{300}$KVKLPL$^{305}$

| # | b'++ | b*++ | b++ | b'+ | b*+ | b+ | seq | y'++ | y*++ | y++ | y'+ | y*+ | y+ | # |
|---|------|------|-----|-----|-----|----|-----|------|------|-----|-----|-----|----|---|
| 1 | — | 56.54 | 65.05 | | 112.08 | 129.10 | K | 588.85 | 589.34 | 597.86 | 1176.70 | 1177.68 | 1194.71 | M |
| 2 | | 106.08 | 114.59 | | 211.14 | 228.17 | V | 524.81 | | 533.81 | 1048.60 | | 1066.61 | 5 |
| 3 | 418.24 | 418.73 | 427.24 | 835.47 | 836.45 | 853.48 | K | 475.27 | — | 484.28 | 949.54 | | 967.55 | 4 |
| 4 | 474.78 | 475.27 | 483.78 | 948.55 | 949.54 | 966.56 | L | 162.62 | | 171.62 | 324.23 | | 342.24 | 3 |
| 5 | 523.31 | 523.80 | 532.31 | 1045.60 | 1046.59 | 1063.61 | P | 106.08 | | 115.08 | 211.14 | | 229.15 | 2 |
| | — | — | — | — | — | — | L | 57.55 | | 66.55 | 114.09 | | 132.10 | 1 |

Figure S4. **The XL-MS results of the ternary complex, Bcp1(K187)-Rkm1(K302).**

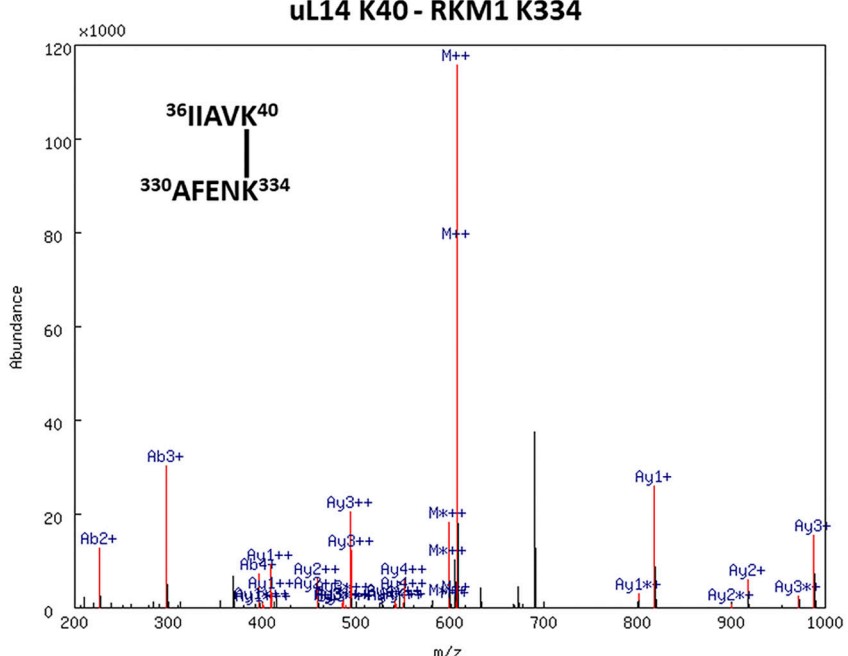

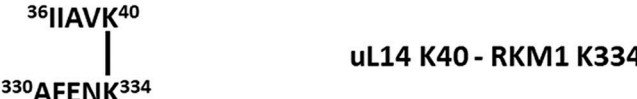

uL14 K40 - RKM1 K334

### Rpl23, ³⁶IIAVK⁴⁰

| # | b'++ | b*++ | b++ | b'+ | b*+ | b+ | seq | y'++ | y*++ | y++ | y'+ | y*+ | y+ | # |
|---|------|------|-----|-----|-----|----|-----|------|------|-----|-----|-----|----|---|
| 1 | — | — | 57.55 | — | — | 114.09 | I | 598.86 | 599.35 | 607.86 | 1196.70 | 1197.69 | 1214.71 | M |
| 2 | | — | 114.09 | | | 227.18 | I | 542.31 | 542.81 | 551.32 | 1083.62 | 1084.60 | 1101.63 | 4 |
| 3 | | — | 149.61 | | | 298.21 | A | 485.77 | 486.26 | 494.78 | 970.54 | 971.52 | 988.55 | 3 |
| 4 | | | 199.14 | | | 397.28 | V | 450.25 | 450.74 | 459.26 | 899.50 | 900.48 | 917.51 | 2 |
| | | | | | | | K | 400.72 | 401.21 | 409.72 | 800.43 | 801.41 | 818.44 | 1 |

### RKM1, ³³⁰AFENK³³⁴

| # | b'++ | b*++ | b++ | b'+ | b*+ | b+ | seq | y'++ | y*++ | y++ | y'+ | y*+ | y+ | # |
|---|------|------|-----|-----|-----|----|-----|------|------|-----|-----|-----|----|---|
| 1 | — | | 36.53 | — | | 72.04 | A | | | | | | | |
| 2 | | — | 110.06 | | | 219.11 | F | 563.34 | 563.83 | 572.34 | 1125.67 | 1126.65 | 1143.68 | 4 |
| 3 | 165.58 | — | 174.58 | 330.14 | — | 348.16 | E | 489.80 | 490.29 | 498.81 | 978.60 | 979.58 | 996.61 | 3 |
| 4 | 222.60 | 223.09 | 231.60 | 444.19 | 445.17 | 462.20 | N | 425.28 | 425.77 | 434.29 | 849.56 | 850.54 | 867.57 | 2 |
| | | | | | | | K | 368.26 | — | 377.27 | 735.51 | — | 753.52 | 1 |

Figure S5. **The XL-MS results of the ternary complex, uL14(K40)-Rkm1(K334).**

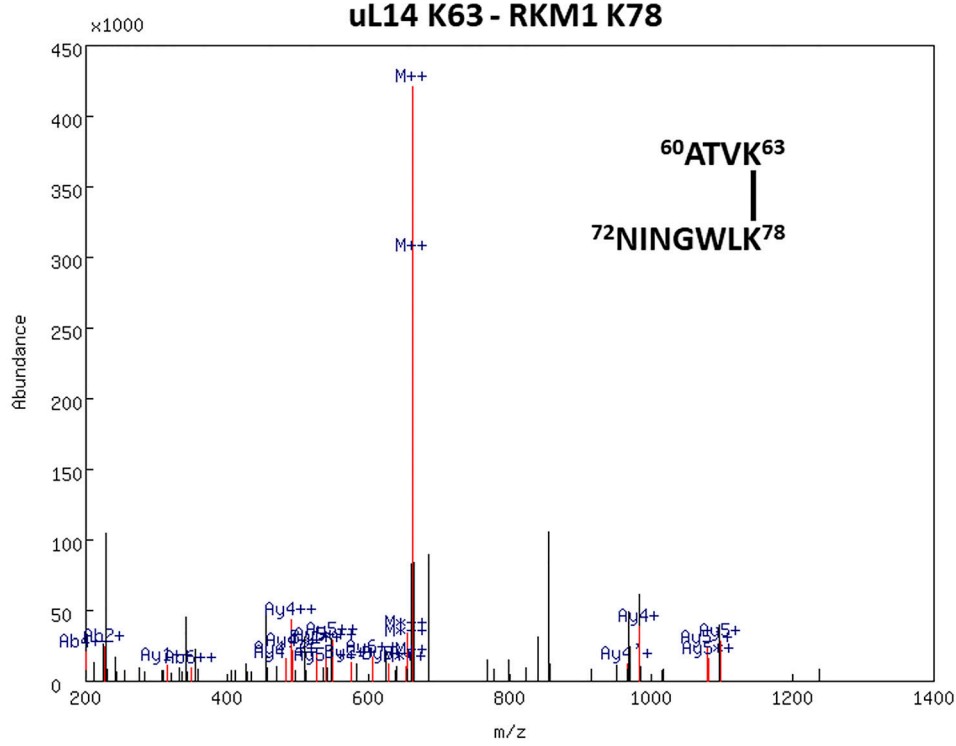

## uL14 K63 - RKM1 K78

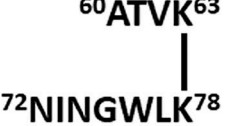

**uL14 K63 - RKM1 K78**

### uL14, ⁶⁰ATVK⁶³

| # | b'++ | b*++ | b++ | b'+ | b*+ | b+ | seq | y'++ | y*++ | y++ | y'+ | y*+ | y+ | # |
|---|---|---|---|---|---|---|---|---|---|---|---|---|---|---|
| 1 | — | — | 36.53 | — | | 72.04 | A | | | | | | | |
| 2 | 78.04 | | 87.05 | 155.08 | | 173.09 | T | 618.86 | 619.35 | 627.86 | 1236.71 | 1237.69 | 1254.72 | 3 |
| 3 | 127.58 | | 136.58 | 254.15 | | 272.16 | V | 568.33 | 568.83 | 577.34 | 1135.66 | 1136.65 | 1153.67 | 2 |
| | | | | | | | K | 518.80 | 519.29 | 527.81 | 1036.59 | 1037.58 | 1054.60 | 1 |

### RKM1, ⁷²NINGWLK⁷⁸

| # | b'++ | b*++ | b++ | b'+ | b*+ | b+ | seq | y'++ | y*++ | y++ | y'+ | y*+ | y+ | # |
|---|---|---|---|---|---|---|---|---|---|---|---|---|---|---|
| 1 | — | 49.52 | 58.03 | — | 98.02 | 115.05 | N | 654.38 | 654.87 | 663.38 | 1307.75 | 1308.73 | 1325.76 | M |
| 2 | — | 106.06 | 114.57 | — | 211.11 | 228.13 | I | 597.36 | 597.85 | 606.36 | 1193.70 | 1194.69 | 1211.71 | 6 |
| 3 | — | 163.08 | 171.59 | — | 325.15 | 342.18 | N | 540.81 | 541.31 | 549.82 | 1080.62 | 1081.60 | 1098.63 | 5 |
| 4 | — | 191.59 | 200.10 | — | 382.17 | 399.20 | G | 483.79 | — | 492.80 | 966.58 | — | 984.59 | 4 |
| 5 | — | 284.63 | 293.14 | — | 568.25 | 585.28 | W | 455.28 | — | 464.29 | 909.56 | — | 927.57 | 3 |
| 6 | — | 341.17 | 349.68 | — | 681.34 | 698.36 | L | 362.24 | — | 371.25 | 723.48 | — | 741.49 | 2 |
| | | | | | | | K | 305.70 | — | 314.71 | 610.39 | | 628.40 | 1 |

Figure S6.   **The XL-MS results of the ternary complex, uL14(K63)-Rkm1(K78).**

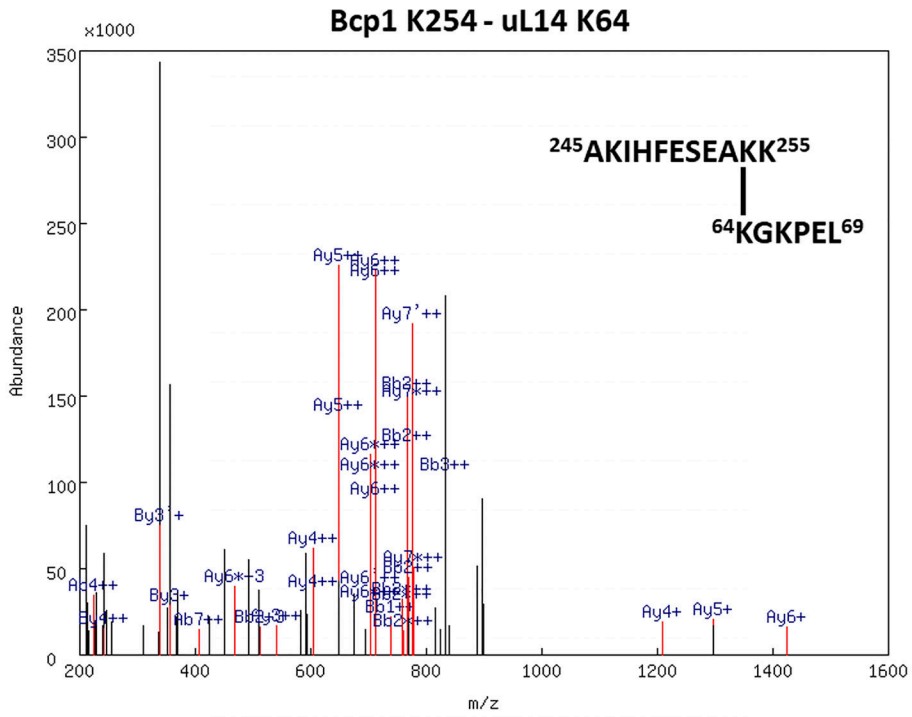

## Bcp1 K254 - uL14 K64

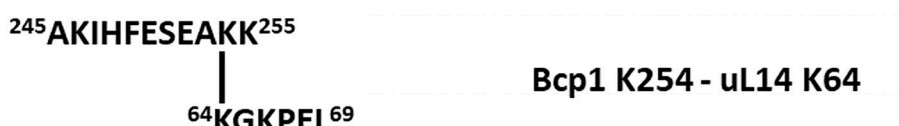

### Bcp1, ²⁴⁵AKIHFESEAKK²⁵⁵

| # | b'+3 | b*+3 | b+3 | b'++ | b++ | b++ | b'+ | b*+ | b+ | seq | y'+3 | y*+3 | y+3 | y'++ | y*++ | y++ | y'+ | y*+ | y+ | # |
|---|---|---|---|---|---|---|---|---|---|---|---|---|---|---|---|---|---|---|---|---|
| 1 | — | — | 24.69 | — | — | 36.53 | | — | 72.04 | A | 668.71 | 669.04 | 674.72 | 1002.57 | 1003.06 | 1011.57 | 2004.13 | 2005.11 | 2022.14 | M |
| 2 | — | 61.71 | 67.38 | — | 92.06 | 100.57 | | 183.11 | 200.14 | K | 645.04 | 645.36 | 651.04 | 967.05 | 967.54 | 976.05 | 1933.09 | 1934.07 | 1951.10 | 10 |
| 3 | — | 99.40 | 105.08 | — | 148.60 | 157.12 | | 296.20 | 313.22 | I | 602.34 | 602.66 | 608.34 | 903.00 | 903.49 | 912.01 | 1805.00 | 1805.98 | 1823.01 | 9 |
| 4 | — | 145.09 | 150.77 | — | 217.13 | 225.64 | | 433.26 | 450.28 | H | 564.64 | 564.97 | 570.65 | 846.46 | 846.95 | 855.46 | 1691.91 | 1692.90 | 1709.92 | 8 |
| 5 | — | 194.11 | 199.79 | — | 290.67 | 299.18 | | 580.32 | 597.35 | F | 518.96 | 519.28 | 524.96 | 777.93 | 778.42 | 786.94 | 1554.85 | 1555.84 | 1572.86 | 7 |
| 6 | 236.80 | 237.13 | 242.80 | 354.70 | 355.19 | 363.70 | 708.38 | 709.37 | 726.39 | E | 469.93 | 470.26 | 475.94 | 704.40 | 704.89 | 713.40 | 1407.78 | 1408.77 | 1425.79 | 6 |
| 7 | 265.81 | 266.14 | 271.81 | 398.21 | 398.70 | 407.22 | 795.41 | 796.40 | 813.43 | S | 426.92 | 427.25 | 432.92 | 639.87 | 640.37 | 648.88 | 1278.74 | 1279.73 | 1296.75 | 5 |
| 8 | 308.82 | 309.15 | 314.83 | 462.73 | 463.22 | 471.74 | 924.46 | 925.44 | 942.47 | E | 397.91 | 398.24 | 403.91 | 596.36 | 596.85 | 605.36 | 1191.71 | 1192.69 | 1209.72 | 4 |
| 9 | 332.50 | 332.83 | 338.51 | 498.25 | 498.74 | 507.26 | 995.49 | 996.48 | 1013.51 | A | 354.89 | 355.22 | 360.90 | 531.84 | 532.33 | 540.84 | 1062.67 | 1063.65 | 1080.68 | 3 |
| 10 | 620.01 | 620.34 | 626.02 | 929.51 | 930.01 | 938.52 | 1858.02 | 1859.01 | 1876.03 | K | 331.21 | 331.54 | 337.22 | 496.32 | 496.81 | 505.32 | 991.63 | 992.61 | 1009.64 | 2 |
| | — | — | — | — | — | — | | | | K | 43.71 | 44.03 | 49.71 | 65.05 | 65.55 | 74.06 | 129.10 | 130.09 | 147.11 | 1 |

### uL14, ⁶⁴KGKPEL⁶⁹

| # | b'+3 | b*+3 | b+3 | b'++ | b*++ | b++ | b'+ | b*+ | b+ | seq | y'+3 | y*+3 | y+3 | y'++ | y*++ | y++ | y'+ | y*+ | y+ | # |
|---|---|---|---|---|---|---|---|---|---|---|---|---|---|---|---|---|---|---|---|---|
| 1 | 487.95 | 488.27 | 493.95 | 731.41 | 731.91 | 740.42 | 1461.82 | 1462.81 | 1479.83 | K | | | | | | | | | | |
| 2 | 506.95 | 507.28 | 512.96 | 759.93 | 760.42 | 768.93 | 1518.84 | 1519.83 | 1536.85 | G | 175.77 | 176.10 | 181.78 | 263.16 | 263.65 | 272.16 | 525.30 | 526.29 | 543.31 | 5 |
| 3 | 549.65 | 549.98 | 555.65 | 823.97 | 824.46 | 832.98 | 1646.94 | 1647.92 | 1664.95 | K | 156.77 | 157.09 | 162.77 | 234.64 | 235.14 | 243.65 | 468.28 | 469.27 | 486.29 | 4 |
| 4 | 582.00 | 582.33 | 588.01 | 872.50 | 872.99 | 881.50 | 1743.99 | 1744.97 | 1762.00 | P | 114.07 | | 120.07 | 170.60 | | 179.60 | 340.19 | | 358.20 | 3 |
| 5 | 625.02 | 625.34 | 631.02 | 937.02 | 937.51 | 946.03 | 1873.03 | 1874.02 | 1891.04 | E | 81.72 | | 87.72 | 122.07 | | 131.08 | 243.13 | | 261.14 | 2 |
| | — | — | — | — | — | — | | | — | L | 38.70 | | 44.71 | 57.55 | | 66.55 | 114.09 | — | 132.10 | 1 |

Figure S7.  **The XL-MS results of the ternary complex, Bcp1(K254)-uL14(K64).**

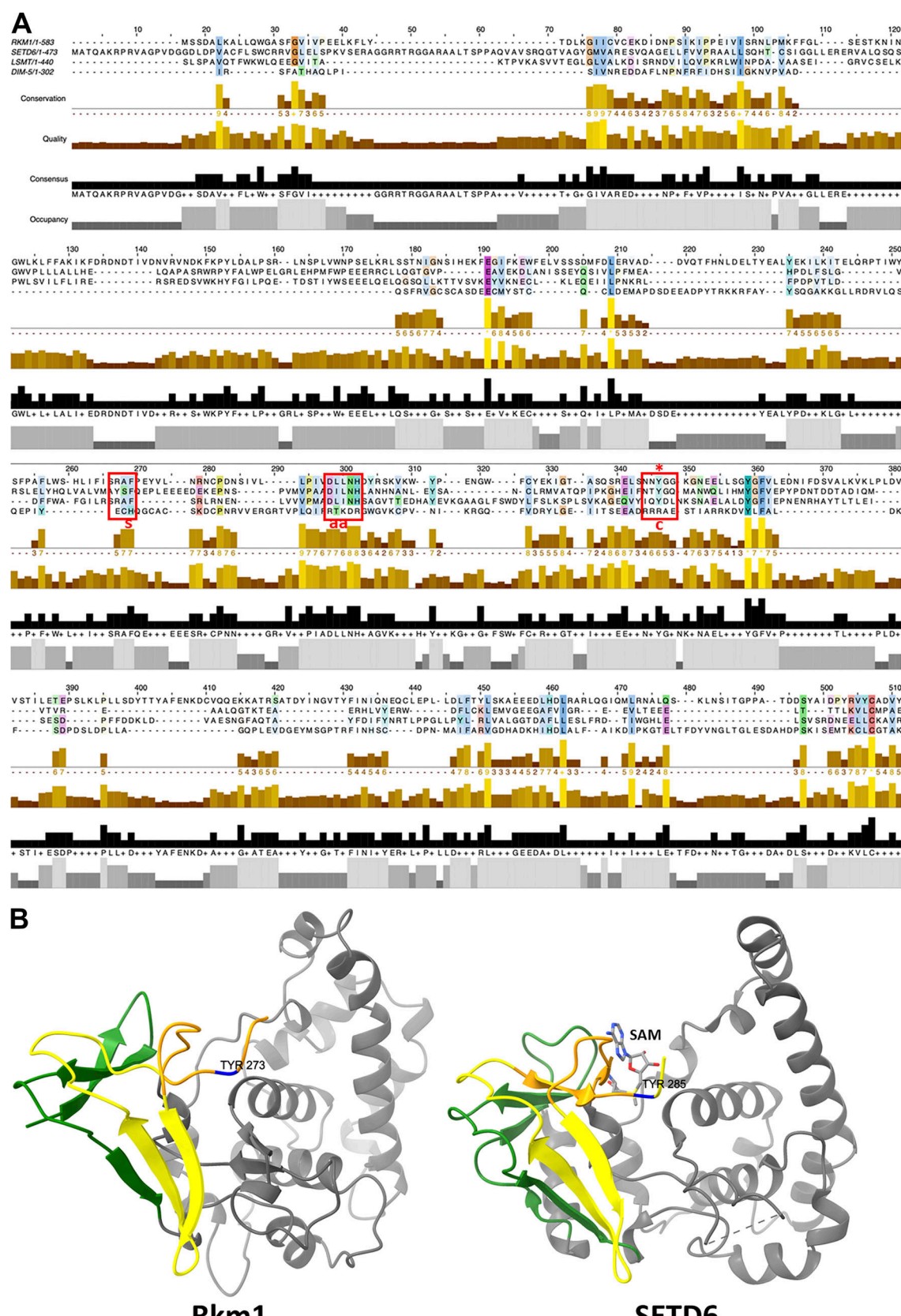

Figure S8. **The sequence alignment and comparison of the SET domain of Rkm1 and other methyltransferases. (A)** The sequence alignment (Jalview Alignment by ClustalW) of Rkm1, *SETD6* (human, GenBank AAH22451), *LSMT* (PDB accession no. 2H2E), and DIM5 (PDB accession no. 1ML9). s: substrate binding; aa: SAM binding; c: catalytic site. **(B)** The structurally conserved core of SET domain is shown in yellow and green, and the insertion region is shown in gray. The active sites, Y273 in Rkm1 and Y285 in SETD6 (3QXY), are shown in blue. The SAM is shown in atom type.

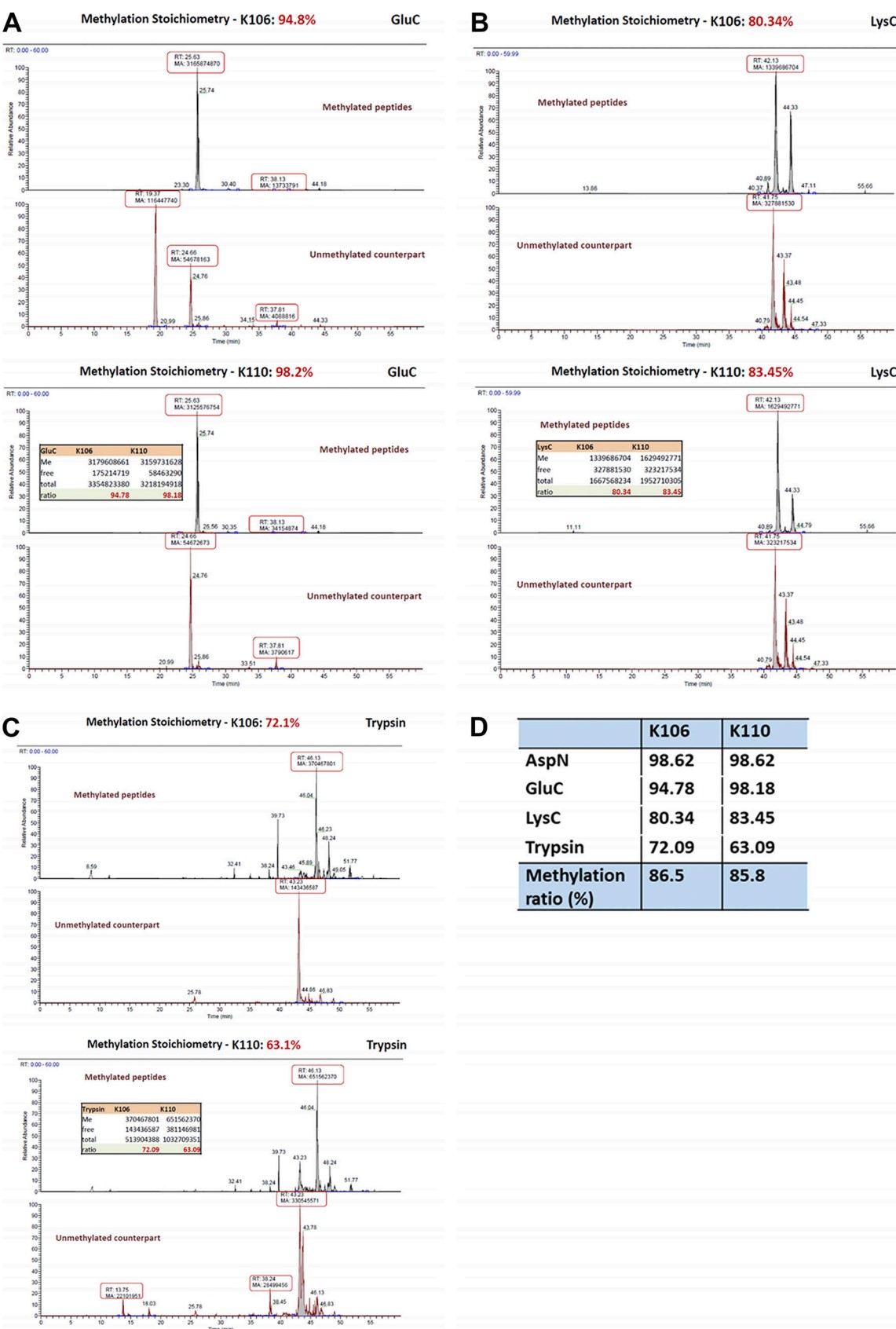

Figure S9. **The MS analysis of methylation sites of uL14 by Rkm1 in the in vitro methylation reaction.** The uL14 was reacted with Rkm1 and SAM in the in vitro methylation reaction. **(A–D)** The proteins were digested with GluC (A), LysC (B), and trypsin (C) and analyzed by mass spectrometry. The ratios of methylated peptides were counted in each reaction (D).

**A**

Ec   1  --------------MIQEQTMLNVADNSGARRVMCIKVLGGSHR----R   31
               .:.....::|.||||||||.:..|.|.|...|         .
Sc   1  MSGNGAQGTKFRISLGLPVGAIMNCADNSGARNLYIIAVKGSGSRLNRLP   50

Ec  32  YAGVGDIIKITIKEAIPRGKVKKGDVLKAVVVRTKKGVRRPDGSVIRFDG   81
             .|.:||::..|:|:..|..:|    |:.|:|||..|..||.||...:|.
Sc  51  AASLGDMVMATVKKGKPELRKK---VMPAIVVRQAKSWRRRDGVFLYFED   97

Ec  82  NACVLLNNNSEQPIGTRIFGPVTRELRSEKFMKIIISLAPEVL         123
            ||.|:.|...|..  |:.|.|||.:|.  :::.:.::.|:..:|:
Sc  98  NAGVIANPKGEMK-GSAITGPVGKEC-ADLWPRVASNSGVVV         137

**B**

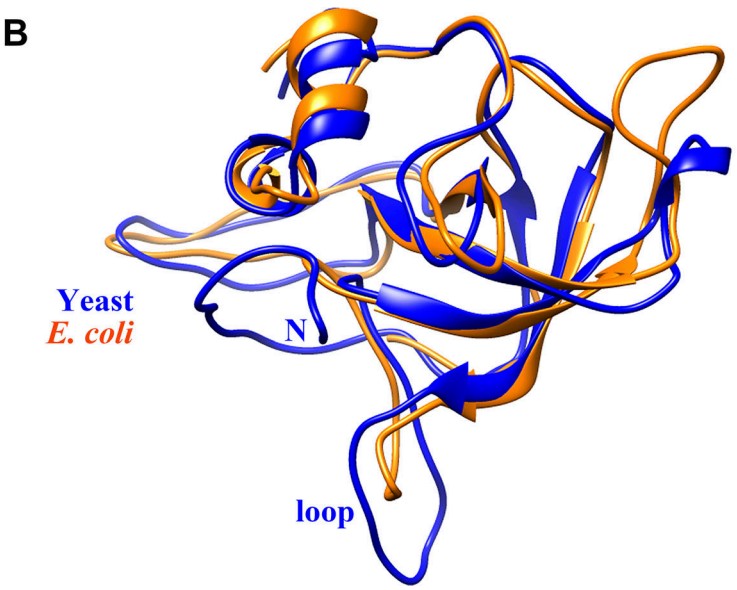

Figure S10.   **Comparisons of *E. coli* and yeast uL14. (A)** The alignment between the amino acid sequences of *E. coli* and yeast uL14 using EMBOSS Needle. **(B)** The structural comparison of *E. coli* (PDB accession no. 1ML5) and yeast (PDB accession no. 5H4P) uL14.

Video 1.   **Molecular dynamics simulation shows that protein dynamics enable the movement of Lys106 and Lys110 residues to the Rkm1 active site for methylation.** Rkm1: Pink; Bcp1: Green; uL14: Blue. The video is related to Fig. 4 G.

**Provided online are Table S1 and Table S2. Table S1 shows yeast strains used in this study. Table S2 lists plasmids used in this study.**

