## [Peer Review File · The Journal of Cell Biology]

Dual protection by Bcp1 and Rkm1 ensures incorporation of uL14 into pre-60S ribosomal subunits

Min-Chi Yeh, Ning-Hsiang Hsu, Hao-Yu Chu, Cheng-Han Yang, Pang-Hung Hsu, Chi-Chi Chou, Jing-Ting Shie, Wei-Ming Lee, Meng-Chiao Ho, and Kai-Yin Lo

Corresponding Author(s): Kai-Yin Lo, National Taiwan University and Meng-Chiao Ho, Institute of Biological Chemistry, Academia Sinica

Review Timeline:

Submission Date:	2023-06-28
Editorial Decision:	2023-08-19
Revision Received:	2024-02-13
Editorial Decision:	2024-03-11
Revision Received:	2024-03-27

Monitoring Editor: Michael Rout

Scientific Editor: Dan Simon

Transaction Report:

DOI: <https://doi.org/10.1083/jcb.202306117>

August 19, 2023

Re: JCB manuscript #202306117

Prof. KAI-YIN LO
National Taiwan University
No. 1, Sec. 4, Roosevelt Rd.
Taipei 10617
Taiwan

Dear Prof. Lo,

Thank you for submitting your manuscript entitled "The dual protection for ribosomal protein uL14 by its dedicated chaperone and methyltransferase." The manuscript has been evaluated by expert reviewers, whose reports are appended below. Unfortunately, after an assessment of the reviewer feedback, our editorial decision is against publication in JCB.

You will see that although the reviewers feel that the work presents intriguing data they also state that additional experiments are necessary in order to provide *in vivo* confirmation for the proposed model that Bcp1 and Rkm1 function to stabilize nascent uL14. Additionally, the reviewers also ask to clarify the physiological relevance of Rkm1 since its absence does not have a significant effect while uL14 is an essential protein. Moreover, there is agreement that the quality of the presented structure is poor, given current capabilities.

Given interest in the topic, we would be open to resubmission to JCB of a significantly revised and extended manuscript that fully addresses all of the reviewers' concerns and is subject to further peer-review. If you would like to resubmit this work to JCB, we ask that you please first submit a revision plan detailing the experiments that you will undertake in order to address the reviewer concerns. Please note that priority and novelty would be reassessed at resubmission of the revised manuscript.

Although your manuscript is intriguing, we feel that the points raised by the reviewers are more substantial than can be addressed in a typical revision period. If you wish to expedite publication of the current data, it may be best to pursue publication at another journal.

Regardless of how you choose to proceed, we hope that the comments below will prove constructive as your work progresses. We would be happy to discuss the reviewer comments further once you've had a chance to consider the points raised in this letter. You can contact the journal office with any questions, cellbio@rockefeller.edu or call (212) 327-8588.

Thank you for thinking of JCB as an appropriate place to publish your work.

Sincerely,

Michael Rout, PhD
Monitoring Editor
Journal of Cell Biology

Dan Simon, PhD
Scientific Editor
Journal of Cell Biology

Reviewer #1 (Comments to the Authors (Required)):

In eukaryotes, the universal ribosomal protein uL14 interacts with two protein partners: (1) Bcp1, the escortin that triggers RanGTP-independent disassembly of a Kap:uL14 complex in the nucleus, and (2) Rkm1, the methyltransferase whose precise role in the ribosome assembly remains unclear. In the manuscript entitled "The dual protection system for ribosomal protein uL14 by its dedicated chaperone and methyltransferase", Lo and co-workers suggest that Rkm1 provides another level of protection to uL14 during its journey from the cytoplasm to the nuclear compartment, and senses the quality of uL14 prior to incorporation.

Comments and suggestions:

(1) The authors convincingly show that Bcp1:uL14:Rkm1 form a stable trimeric complex, and that uL14 bridges interaction between Bcp1 and Rkm1, both *in vitro* and *in vivo*.

- (2) Rkm1 binds to the nascent uL14 in the cytoplasm; this somehow seems to improve the recognition by the nuclear import machinery. The underlying reason is unclear, however, it is unclear whether this is relevant in vivo.
- (3) Growth of bcp1-ts mutant cells is exacerbated Rkm1 deletion, although for this reviewer the synergistic effect is very mild, to conclude that Rkm1 provides an alternative pathway to protect uL14. Moreover, over-expression of Rkm1 does not rescue bcp1-ts mutant growth defects. uL14 is an essential protein, hence one expects synthetic lethality. Note that rkm1 Δ cells do not exhibit growth defects.
- (4) uL14 protein levels are not affected in rkm1 Δ cells. Only in the bcp1-ts mutant background uL14 levels are affected. Is this specific to uL14? Here it would be important to test protein levels of another ribosomal protein (s), instead of using another ribosome assembly factor Puf6.
- (5) They show that the trimeric complex is transported into the nucleus where it is released from the import machinery. They reveal a loop region within uL14 that Bcp1 recognizes to trigger the disassembly of the Kap:uL14:Rkm1 complex. Consistent with this, the disassembly of the Kap:uL14 Δ loop:Rkm1 complex by Bcp1 was affected. Can the authors strengthen this finding in vivo for e.g. by performing an IP to show that the Rkm1 co-enriches the importin in the uL14 Δ loop mutant?
- (6) How is Rkm1 released from uL14 prior to incorporation into the 60S pre-ribosome? Here the authors claim, through in vitro assays, that it is the methylation of uL14. Surprisingly, only methylated uL14 is released from Rkm1 by Bcp1 in vitro. Can the authors gather evidence that this takes place in vivo i. e. is uL14 isolated via Bcp1-myc not methylated, since it forms a stable trimeric complex in vivo?
- (7) How does Bcp1 sense the methylation status of uL14 to trigger dissociation? How does this occur in rkm1 Δ cells, or the uL14 mutants where methylation is (or should be) absent? Would over-expression of a catalytic-dead Rkm1 be toxic in a bcp1-ts mutant?
- (8) The novel aspect of this manuscript is the claim that the quality of uL14 is tested by Rkm1 prior to its release by Bcp1 for incorporation, and this is somehow intertwined with methylation activity of Rkm1. The conclusions for this aspect are supported primarily by in vitro approaches, but no in vivo evidence is provided. Is a uL14 Δ loop mutant which binds to Rkm1 unmethylated in vivo? What are the in vivo levels of uL14 Δ loop?
- (9) A direct prediction from the model is Rkm1 shuttles between the nucleus and cytoplasm. How is Rkm1 exported into the cytoplasm to engage in another round of uL14 import? Can the authors shed light on this?
- (10) The authors probe the organization of the trimeric complex by cryo-EM and XL-MS. The model of the complex is far from convincing especially at 5.8 Angstrom resolution. The authors should work on improving the resolution.

Reviewer #2 (Comments to the Authors (Required)):

This manuscript describes interplay among the methyltransferase Rkm1, the escortin Bcp1, karyopherins Kap122 and Kap123, and ribosomal protein uL14, to escort uL14 into the nucleus and into nascent ribosomes, in yeast.

The major results:

- (1) These proteins form a complex ...reasonable data
- (2) Rkm1 accompanies uL14 into the nucleus...., yes, maybe... they show poor quality micrographs showing that Rkm1 requires uL14 to get into the nucleus, but does this prove the point that they accompany each other, beyond their previous demonstration that Bcp1 interacts with uL14 in the nucleus. Can uL14 enter the nucleus in the bcp1 conditional lethal (Ts-)mutant(maybe this was previously shown?)
- (3) They use low resolution cryo-EM, crosslinking mass spec, and modeling to build a structure of the complex...is the high resolution model valid based on the data?
- (4) Interaction of Bcp1 with uL14 is necessary to release uL14 from Kap...ok but the authors state that "Rkm1 and uL14 first interact in the cytoplasm and are co-transported into the nucleus". Where did they show that they first interact in the cytoplasm?
- (5) Bcp1 triggers release of Rkm1 from methylated uL14 ...ok
- (6) Disassembly of Bcp1, Rkm1, and uL14 requires Bcp1 and methylation of uL14 ...ok

To improve this manuscript, I urge that authors to address the following concerns:

- (1)The manuscript is extremely poorly written and therefore difficult to understand. Can the authors enlist the aid of someone who is a native speaker of English to clean up the many grammatical error? However, more importantly, I am concerned that the rationale for performing the experiments and for interpretations of the experiments simply is not made clear enough. Figure legends and descriptions in Results need to be augmented.
- (2) The story seems incomplete.
 - (a)The title, Abstract and Introduction emphasize that Rkm1 and Bcp1 "protect" uL14, yet I believe that this idea needs to be examined in much more depth. Yes, the authors knew before or showed here that these proteins form complexes with uL14, and that uL14 is unstable in their absence. They use cryo-EM to obtain a very low resolution structure and cross-linking and modeling to create a higher resolution structure of Bcp1, Rkm1, and uL14. However, I suggest that the authors carry out a limited proteolysis assay of uL14 in the presence of different combinations of the other proteins to more directly test their hypothesis. Have the authors made a strong enough argument that Bcp1 and Rkm1 directly protect the internal loop of uL14?

Perhaps someone more expert in modeling could help me here?

(3) The authors fail to justify why understanding the function of Rkm1 is important. The authors describe how methylation of uL14 by Rkm1 triggers release of methylated uL14 from Rkm1 by Bcp1. Yet, Rkm1 is not essential.

Therefore, the authors need to reconcile in much more detail why Rkm1 is "not essential". They state that there is a slight underaccumulation of 60S ribosomal subunits in the rkm11 knockout. Is there no effect at all on cell growth rate? Help us understand how to appreciate why Rkm1 is important. For example, the authors should explore how and to what extent uL14 is assembled into ribosomes without Rkm1. I think that uL14 is essential, so its assembly into ribosomes must be essential.

(4) I am not certain about the state of the art of ribosomal protein escortins, but I was disappointed that there is no information in this manuscript describing exactly how and when Bcp1 enables insertion of uL14 into nascent ribosomes.

(5) To make this manuscript acceptable to the broad readership of JCB, the authors need to completely rebuild the Introduction. Exactly what do we know about how escortins enable transport and more importantly, installation of ribosomal proteins into ribosomes? What are the gaps in our knowledge that the work described in this manuscript fill?

Some minor issues to address:

Figure 1B: What proteins are in each OD peak? This needs more labeling.

Figure 1C: The gel seems distorted; the Bcp1-myc bands migrate more slowly in lanes 4 and 5 than in the other lanes, while the uL14 bands migrate faster???

Figure 2A Please repeat; the micrographs are of poor quality; the GFP signals are faint.

Figure 2B Please label as GST-KAP122 and GST-KAP123

Figure 2D rather than labeling as TAP, label as Rkm1-TAP. Why does uL14 migrate faster than in other lanes?

Figure 3B Please label the N and C termini of the proteins to help us orient better.

Figure 4A It is difficult to see the portion of L14 embedded in the ribosome...brighter color maybe?

Figure 5A, left Does this mean that aa1-36 are flexible but aa37-52 are disordered but not flexible?

Figure 5e The rationale for these experiments was not made clear enough.

and 5E TOP panel: can the authors show these as westerns maybe more clear

Figure 6A It would be easier to understand this complex figure if the cartoon was at the top..the first thing to which one's eyes are drawn

Figure 6B needs more explanation...do the numbers really add up? Why are there equal amounts of Bcp1 detected by western?

Reviewer #3 (Comments to the Authors (Required)):

Yeh et al, J Cell Bio 202306117

on this manuscript Yeh and colleagues use a combination of genetic, biochemical, fluorescence and structural approaches to dissect the structure and function of the Bcp1, Rkm1 and uL14 complex. The experiments are well executed and mostly support the conclusions on the paper. The structure of the complex is not up to current standard in the field however, and overall, it is a borderline candidate for JCB as its relevance for a broader audience is somewhat tenuous.

Major comments:

- It is not true that the ternary complex of Bcp1, Rkm1 and uL14 is too small for cryo-EM. The complex is over 120 kDa, well within the range of current cryo-EM capabilities. As such the cryo-EM data are not current standard. The data claims a resolution of 5.9A, but the map looks significantly worse, and there is no FSC curve, and it is not noted which criteria for this resolution was used.
- The authors must either obtain a much better-quality structure or probe the suggested placement of residues more extensively by making mutants at each of the protein-protein interfaces. These will then also be useful in functional studies (see below).
- Figure 2 needs an input as well as the elution. Without these it is impossible to judge how well this experiment works.
- If Bcp1 really dissociates the Rkm1/uL14 complex, then one would expect that a Bcp1 mutant that fails to interact with uL14 and Rkm1 should be deleterious, and that its defect should be suppressable by a mutation in Rkm1 that weakens the binding to uL14
- The IF data do not really support the model for how the process works. The data indicate that Rkm1 depends on uL14 for

nuclear import, and that Bcp1 depends on binding Kap for nuclear import. This latter piece of data is not consistent with the model that Bcp1 releases uL14 (or the uL14 Rkm1 complex) from karyopherins in the nucleus.

- The finding that Bcp1-dependent release of Rkm1 from uL14 requires its methylation is very interesting. However, it is NOT supportive of the conclusion that this is a QC step for uL14 functionality. It merely says that it is a QC step for uL14 methylation. To show that it monitors uL14 integrity, the authors would need to show that a misfolded mutant cannot be released. I will let the authors and the editor decide whether it is appropriate to just tone down the conclusion or carry out this other experiment.

Minor comments:

- There are 46 proteins in the yeast 60S subunit, not 39 as claimed in the introduction.
- The manuscript could benefit from some English language editing
- P.13 has uL14 as ul14.
- Figures 1 and 2 refer to uL14 alternately as uL14 or L23. The authors need to pick one nomenclature.

Dear Editor,

We appreciate the opportunity to revise our manuscript. In response to the reviewers' comments, we have conducted additional experiments and performed another round of cryoEM. In this revision, we collected a new dataset comprising 5484 micrographs, which contain 1,931,871 particles of cross-linked Bcp1/uL14/Rkm1 complex. However, the reported crystal structures revealed the long flexible N-terminus loop of Bcp1 and uL14 also contains long disordered loops, which require a chaperone to stabilize it. In our simulation data, we observed dynamic conformation changes of Bcp1-Rkm1- uL14 heterotrimer. Despite our efforts, we still could not obtain a high-resolution cryo-EM structure due to the flexible nature of the Bcp1-Rkm1-uL14 heterotrimer. Therefore, to probe the trimeric structure, we used simulation docking guided by XL-MS and confirmed that the overall envelope of docking result is consistent with the cryo-EM map. Our result reveals the spatial correlation between Bcp1, Rkm1, and uL14, demonstrating the protection model.

Please understand that the structural data is just part of the evidence to support our model. We have conducted numerous experiments to support the conclusions. Additionally, we have shown that Rkm1 acts as another chaperone of uL14, with its methylation activity regulated by Bcp1, serving as an additional surveillance point for uL14 quality.

Furthermore, we have thoroughly rewritten the introduction and discussion sections to meet the standards of JCB. We are confident that the quality of our work has significantly improved in this revised version. We trust that the revisions adequately address the concerns raised by the reviewers, making the manuscript suitable for publication. We are grateful for your consideration and assistance in advancing this manuscript.

Thank you sincerely for your time and support.

Best regards,

Kai-Yin

Professor, Department of Agricultural Chemistry
National Taiwan University

Meng-Chiao Ho

Professor, Institute of Biological Chemistry
Academia Sinica

Reviewer #1 (Comments to the Authors (Required)):

In eukaryotes, the universal ribosomal protein uL14 interacts with two protein partners: (1) Bcp1, the escortin that triggers RanGTP-independent disassembly of a Kap:uL14 complex in the nucleus, and (2) Rkm1, the methyltransferase whose precise role in the ribosome assembly remains unclear. In the manuscript entitled "The dual protection system for ribosomal protein uL14 by its dedicated chaperone and methyltransferase", Lo and co-workers suggest that Rkm1 provides another level of protection to uL14 during its journey from the cytoplasm to the nuclear compartment, and senses the quality of uL14 prior to incorporation.

Comments and suggestions:

(1) The authors convincingly show that Bcp1:uL14:Rkm1 form a stable trimeric complex, and that uL14 bridges interaction between Bcp1 and Rkm1, both in vitro and in vivo.

(2) Rkm1 binds to the nascent uL14 in the cytoplasm; this somehow seems to improve the recognition by the nuclear import machinery. The underlying reason is unclear, however, it is unclear whether this is relevant in vivo.

According to NLS prediction (Ba et al., 2009), Rkm1 lacks an NLS, while uL14 contains one. This aligns with our interaction results, where Kap121 and Kap123 can interact with uL14 but not with Rkm1. However, when Rkm1 and uL14 form a complex, Kap121/Kap123 can interact with the Rkm1/uL14 complex. Additionally, the nuclear localization of Rkm1 was abolished when uL14 synthesis was halted. These findings suggest that the import of Rkm1 relies on uL14. In our investigation, we observed that the interaction with Rkm1 during the transport pathway aids in stabilizing uL14. We made the description more clearly in the revision.

Ba, A.N.N., A. Pogoutse, N. Provar, and A.M. Moses. 2009. NLStradamus: a simple Hidden Markov Model for nuclear localization signal prediction. *BMC Bioinformatics*. 10:202

(3) Growth of bcp1-ts mutant cells is exacerbated Rkm1 deletion, although for this reviewer the synergistic effect is very mild, to conclude that Rkm1 provides an alternative pathway to protect uL14. Moreover, over-expression of Rkm1 does not rescue bcp1-ts mutant growth defects. uL14 is an essential protein, hence one expects synthetic lethality. Note that rkm1 Δ cells do not exhibit growth defects.

Thanks for the suggestion. Typically, karyopherins play a protective role for ribosomal proteins during the import process (Pillet, 2017). Consequently, Rkm1 redundancy in the protective function is evident only when combined with another mutant within the same pathway.

Pillet, B., V. Mitterer, D. Kressler, and B. Pertschy. 2017. Hold on to your friends: Dedicated chaperones of ribosomal proteins: Dedicated chaperones mediate the safe transfer of ribosomal proteins to their site of pre-ribosome incorporation. *BioEssays*. 39:1-12.

(4) uL14 protein levels are not affected in *rkm1* Δ cells. Only in the *bcp1-ts* mutant background uL14 levels are affected. Is this specific to uL14? Here it would be important to test protein levels of another ribosomal protein (s), instead of using another ribosome assembly factor Puf6.

Several ribosomal proteins have been detected under the same conditions and only uL14 was decreased significantly. These results have been included in the revision in Fig 1E.

(5) They show that the trimeric complex is transported into the nucleus where it is released from the import machinery. They reveal a loop region within uL14 that Bcp1 recognizes to trigger the disassembly of the Kap:uL14:Rkm1 complex. Consistent with this, the disassembly of the Kap:uL14 Δ loop:Rkm1 complex by Bcp1 was affected. Can the authors strengthen this finding in vivo for e.g. by performing an IP to show that the Rkm1 co-enriches the importin in the uL14 Δ loop mutant?

We conducted immunoprecipitation experiments to demonstrate this point. When Kap121/Kap123 were immuno-purified, the interaction with Rkm1 was enhanced in the uL14 Δ loop strain. This data is consistent with the in vitro study, affirming that the direct interaction between Bcp1 and uL14 is required for the release of Rkm1/uL14 complex from Kap. The data has been included as Fig 5G in the revision.

(6) How is Rkm1 released from uL14 prior to incorporation into the 60S pre-ribosome? Here the authors claim, through in vitro assays, that it is the methylation of uL14. Surprisingly, only methylated uL14 is released from Rkm1 by Bcp1 in vitro. Can the authors gather evidence that this takes place in vivo i.e. is uL14 isolated via Bcp1-myc not methylated, since it forms a stable trimeric complex in vivo?

We conducted immunoprecipitation experiments to elucidate this. uL14 and uL14(RR) were immuno-purified, revealing that uL14(RR) exhibited higher signals for Bcp1 and Rkm1 interactions. Similarly, Rkm1(Y273F) also showed stronger interactions with Bcp1 and uL14. These findings demonstrate that uL14(RR), with mutations at the methylation sites, and Rkm1(Y273F), with mutations at the methyltransferase site, were retained in the form of a ternary complex. The corresponding data has been included as Fig 6F and 6G in the revised manuscript.

(7) How does Bcp1 sense the methylation status of uL14 to trigger dissociation? How does this occur in *rkm1* Δ cells, or the uL14 mutants where methylation is (or should be) absent? Would over-expression of a catalytic-dead Rkm1 be toxic in a *bcp1-ts* mutant?

It is possible that the interaction between Rkm1 and uL14, or methylated-uL14, is different. Consequently, Bcp1 might exclusively dissociate methylated-uL14 from Rkm1. In *rkm1* Δ , uL14 remains unmethylated and could be released from Kap by Bcp1 (Ting et al., 2017). To further support this data, growth tests were conducted with the *bcp1ts* mutant in the presence of overexpressed *rkm1(Y273F)* or *uL14(RR)* mutants, revealing a slower growth phenotype for *bcp1ts* in the presence of these mutants. The data has been included as Fig 6H and 6I in the revision.

Ting, Y.H., T.J. Lu, A.W. Johnson, J.T. Shie, B.R. Chen, S.S. Kumar, and K.Y. Lo. 2017. Bcp1 is the Nuclear Chaperone of the 60S ribosomal protein Rpl23 in *Saccharomyces cerevisiae*. *J Biol Chem.* 292:585-596.

(8) The novel aspect of this manuscript is the claim that the quality of uL14 is tested by Rkm1 prior to its release by Bcp1 for incorporation, and this is somehow intertwined with methylation activity of Rkm1. The conclusions for this aspect are supported primarily by in vitro approaches, but no in vivo evidence is provided. Is a uL14 Δ loop mutant which binds to Rkm1 unmethylated in vivo? What are the in vivo levels of uL14 Δ loop?

We attempted to immunopurify uL14 Δ loop and detect the methylation signal using an anti-Lys(methylation) antibody. Unfortunately, we were unable to detect signals even in the wild-type (WT). This may be due to the possibility that not all uL14 proteins were methylated in vivo, adding complexity to the ribosome. Alternatively, the presence of a demethylase may render methylations dynamic in vivo. To explore this concept, we utilized uL14(RR) and *rkm1(Y273F)* mutants (Fig 6F, 6G) to provide support for the notion that the disassembly of the ternary complex depends on the methylation status of uL14.

(9) A direct prediction from the model is Rkm1 shuttles between the nucleus and cytoplasm. How is Rkm1 exported into the cytoplasm to engage in another round of uL14 import? Can the authors shed light on this?

To further demonstrate that Rkm1 is a shuttling protein, its cellular distribution was examined in the *kap121ts*, *kap123* Δ , and *crm1T539C* mutants. The nuclear signals of Rkm1 became diffused in the cytoplasm in the *kap123* Δ mutants (Fig 2C). Rkm1

became more concentrated in the nucleus in *crm1T539C* strain when the inhibitor leptomycin B was included (**Fig 2D**). The above data suggests that the import of Rkm1 depends majorly on Kap123 and is exported via Crm1 after its release from uL14.

(10) The authors probe the organization of the trimeric complex by cryo-EM and XL-MS. The model of the complex is far from convincing especially at 5.8 Angstrom resolution. The authors should work on improving the resolution.

Thanks for reviewer's comments. We understood that the resolution is far from convincing at 5.8 Å, and we purposefully removed the resolution number in the main text and the FSC curve in the figure. We are sorry that the 5.8 Å was mistakenly left in the figure legend. Previous data was collected on a small dataset (382 micrographs) on Titan Krios. During this revision, we collected a new dataset of 5484 micrographs containing 1,931,871 particles of cross-linked Bcp1/uL14/Rkm1 complex. However, the reported crystal structures revealed the long flexible N-terminus loop of Bcp1 and uL14 also contains long disordered loops, which require a chaperone to stabilize it. In our simulation data, we observe dynamic conformation changes of Bcp1-Rkm1- uL14 heterotrimer. Therefore, despite our efforts, we still could not obtain a high-resolution cryo-EM structure due to the flexible nature of the Bcp1-Rkm1-uL14 heterotrimer. Despite the fact that we could not observe any structure features of alpha-helices, indicating a low-resolution map, the FSC approach indicating ~5 Å resolution seems to overestimate our map resolution in this case, so we used d99 approach, which estimates the resolution related to map details in the real space (Afonine et al., 2018). d99 is the resolution at which a map gradually removed the highest resolution shell starts to differ from the original map. The resolution of the map is estimated to be around 16.32 Å, consisting with our manual observation of the map quality. In the revised manuscript, we have updated the estimated resolution from d99 in the figure S2 and in the method.

Therefore, to probe trimeric structure, we use simulation docking guided by XL-MS and confirm that the overall envelope of docking result is consistent with the cryo-EM map. Our result reveals the spatial correlation between Bcp1, Rkm1, and uL14, demonstrating the protection model.

Afonine, P.V., B.P. Klaholz, N.W. Moriarty, B.K. Poon, O.V. Sobolev, T.C.Terwilliger, P.D. Adams, and A. Urzhumtsev. 2018. New tools for the analysis and validation of cryo-EM maps and atomic models. *Acta Crystallogr D Struct Biol.* 74:814-840.

Reviewer #2 (Comments to the Authors (Required)):

This manuscript describes interplay among the methyltransferase Rkm1, the escortin Bcp1, karyopherins Kap122 and Kap123, and ribosomal protein uL14, to escort uL14 into the nucleus and into nascent ribosomes, in yeast.

The major results:

(1) These proteins form a complex ...reasonable data

(2) Rkm1 accompanies uL14 into the nucleus....., yes, maybe... they show poor quality micrographs showing that Rkm1 requires uL14 to get into the nucleus, but does this prove the point that they accompany each other, beyond their previous demonstration that Bcp1 interacts with uL14 in the nucleus. Can uL14 enter the nucleus in the bcp1 conditional lethal (Ts-)mutant(maybe this was previously shown?)

Thanks for the suggestion. The import of uL14 does not depend on Bcp1 (Fig S1F). We provided two pieces of data to support this point. The nuclear localization of Rkm1 was lost when uL14 was depleted (Fig 2A). Rkm1 alone did not show interaction with Kap in vitro, but the interaction was enhanced by the presence of uL14 (Fig 2B). The description has been revised for better clarity in the revision and Fig 2A has been replaced with a higher-quality one.

(3) They use low resolution cryo-EM, crosslinking mass spec, and modeling to build a structure of the complex...is the high resolution model valid based on the data?

Our high-resolution docking model is based on the alpha fold predicted structure and crosslinking mass result. The docking model fits into the low-resolution cryo-EM, which provides some basis of validation. As we showed in the Fig 4F and video 1, the Bcp1-Rkm1-uL14 trimer is dynamic, allowing the methylation of various lysine residues, but, unfortunately, the dynamic nature prevents the high-resolution cryo-EM determination.

Our model suggests that uL14 is sandwiched by Bcp1 and Rkm1. More importantly, uL14 interacts with Bcp1 by different interface compared to the uL14-Rkm1 interface. This result is supported by the limited proteolysis data (Fig 1F), suggested by the reviewer.

(4)Interaction of Bcp1 with uL14 is necessary to release uL14 from Kap...ok but the authors state that "Rkm1 and uL14 first interact in the cytoplasm and are co-transported into the nucleus". Where did they show that they first interact in the cytoplasm?

According to NLS prediction (Ba et al., 2009), Rkm1 lacks an NLS, while uL14 contains one. This aligns with our interaction results, where Kap121 and Kap123 can interact with uL14 but not with Rkm1. However, when Rkm1 and uL14 form a complex, Kap121/Kap123 can interact with the Rkm1/uL14 complex. Additionally, the nuclear localization of Rkm1 was abolished when uL14 synthesis was halted. These findings suggest that the import of Rkm1 relies on uL14. In our investigation, we observed that the interaction with Rkm1 during the transport pathway aids in stabilizing uL14. We made the description more clearly in the revision.

Ba, A.N.N., A. Pogoutse, N. Provar, and A.M. Moses. 2009. NLStradamus: a simple Hidden Markov Model for nuclear localization signal prediction. *Bmc Bioinformatics*. 10: 202

(5) Bcp1 triggers release of Rkm1 from methylated uL14 ...ok

(6) Disassembly of Bcp1, Rkm1, and uL14 requires Bcp1 and methylation of uL14 ...ok

To improve this manuscript, I urge that authors to address the following concerns:

(1) The manuscript is extremely poorly written and therefore difficult to understand. Can the authors enlist the aid of someone who is a native speaker of English to clean up the many grammatical error? However, more importantly, I am concerned that the rationale for performing the experiments and for interpretations of the experiments simply is not made clear enough. Figure legends and descriptions in Results need to be augmented.

Thanks for the suggestion. The manuscript has been corrected and revised accordingly.

(2) The story seems incomplete.

(a)The title, Abstract and Introduction emphasize that Rkm1 and Bcp1 "protect" uL14, yet I believe that this idea needs to be examined in much more depth. Yes, the authors knew before or showed here that these proteins form complexes with uL14, and that uL14 is unstable in their absence. They use cryo-EM to obtain a very low resolution structure and cross-linking and modeling to create a higher resolution structure of Bcp1, Rkm1, and uL14. However, I suggest that the authors carry out a limited proteolysis assay of uL14 in the presence of different combinations of the other proteins to more directly test their hypothesis. Have the authors made a strong enough argument that Bcp1 and Rkm1 directly protect the internal loop of uL14? Perhaps someone more expert in modeling could help me here?

We did a limited proteolysis experiment (Fig 1F) to demonstrate the protection roles of Rkm1 and Bcp1. Varying amounts of trypsin were introduced to purified uL14, as well as to the purified complexes of Rkm1/uL14, Bcp1/uL14, and Rkm1/Bcp1/uL14. The stability of uL14 was then compared across these conditions. uL14 exhibited the highest sensitivity to trypsin, but the addition of Rkm1 or Bcp1 conferred protection against proteolysis, with Bcp1 demonstrating superior protective capability compared to Rkm1. Remarkably, the presence of both Bcp1 and Rkm1 significantly enhanced the stability of uL14 (Fig 1F). These findings underscore the essential role of the collaborative action between Bcp1 and Rkm1 in stabilizing nascent uL14.

(3) The authors fail to justify why understanding the function of Rkm1 is important. The authors describe how methylation of uL14 by Rkm1 triggers release of methylated uL14 from Rkm1 by Bcp1. Yet, Rkm1 is not essential.

Therefore, the authors need to reconcile in much more detail why Rkm1 is "not essential". They state that there is a slight underaccumulation of 60S ribosomal subunits in the *rkm1* knockout. Is there no effect at all on cell growth rate? Help us understand how to appreciate why Rkm1 is important. For example, the authors should explore how and to what extent uL14 is assembled into ribosomes without Rkm1. I think that uL14 is essential, so its assembly into ribosomes must be essential.

We conducted growth tests *rkm1Δ* under several conditions and did not see growth defects of *rkm1Δ* (Fig S1A). Typically, karyopherins play a protective role in the import process of ribosomal proteins (Pillet, 2017). uL14 is released from Kaps and protected by Bcp1 in the nucleus. Consequently, Rkm1 appears redundant in this protective role, and defects become evident only when coupled with another mutant within the same pathway. Indeed, *rkm1Δ* exhibited synthetic growth defects when combined with *bcp1ts* (Fig 1A), and overexpression of a catalytic-dead Rkm1 mutant, *rkm1Y273F*, was toxic to *bcp1ts* cells (Fig 6H). Moreover, nascent uL14 levels were even lower in the *bcp1ts*rkm1Δ* mutant. These findings affirm that Rkm1 indeed plays a crucial role in uL14 protection.

Pillet, B., V. Mitterer, D. Kressler, and B. Pertschy. 2017. Hold on to your friends: Dedicated chaperones of ribosomal proteins: Dedicated chaperones mediate the safe transfer of ribosomal proteins to their site of pre-ribosome incorporation. *BioEssays*. 39:1-12.

(4) I am not certain about the state of the art of ribosomal protein escortins, but I was disappointed that there is no information in this manuscript describing exactly how

and when Bcp1 enables insertion of uL14 into nascent ribosomes.

We now included more details and cite more references in the discussion.

(5) To make this manuscript acceptable to the broad readership of JCB, the authors need to completely rebuild the Introduction. Exactly what do we know about how escortins enable transport and more importantly, installation of ribosomal proteins into ribosomes? What are the gaps in our knowledge that the work described in this manuscript fill?

Thanks for the suggestion. The introduction has been revised and more information about the escortins have been included.

Some minor issues to address:

Figure 1B: What proteins are in each OD peak? This needs more labeling.

The figure has been relabeled in the revision.

Figure 1C: The gel seems distorted; the Bcp1-myc bands migrate more slowly in lanes 4 and 5 than in the other lanes, while the uL14 bands migrate faster???

The migration is correlated with protein abundance in the gel. uL14 is abundant in the WCE but less in the IP, so the protein band migrates faster in the WCE samples.

Figure 2A Please repeat; the micrographs are of poor quality; the GFP signals are faint.

The image has been replaced in the revision.

Figure 2B Please label as GST-KAP122 and GST-KAP123

The figure has been relabeled in the revision.

Figure 2D rather than labeling as TAP, label as Rkm1-TAP. Why does uL14 migrate faster than in other lanes?

The figure has been relabeled in the revision. The signal at lane 1 is not uL14 because the band did not appear at the correct position.

Figure 3B Please label the N and C termini of the proteins to help us orient better.

The figure has been relabeled in the revision.

Figure 4A It is difficult to see the portion of L14 embedded in the ribosome...brighter color maybe?

The figure has been replaced in the revision.

Figure 5A, left Does this mean that aa1-36 are flexible but aa37-52 are disordered but not flexible?

AA1-36 might be flexible and could not be resolved in the X-ray structure. AA37-52 is also unstructured region but could be resolved in the X-ray.

Figure 5e The rationale for these experiments was not made clear enough.

and 5E TOP panel: can the authors show these as westerns maybe more clear

We revised the paragraph to make the rationale clearer. The signals of GST and GST-Kap121 are very strong in the Coomassive Blue gel. The signals of Bcp1 and uL14 have been shown with Western blotting. The figures were relabeled for better clearance.

Figure 6A It would be easier to understand this complex figure if the cartoon was at the top. the first thing to which one's eyes are drawn

The cartoon is moved to the top in the revision.

Figure 6B needs more explanation...do the numbers really add up? Why are there equal amounts of Bcp1 detected by western?

Although the amount of Bcp1 was more in the lane 8 than lane 7 in Fig 6A, the NTA beads we added was limiting. Thus equal amount of Bcp1 was observed in the Western blotting. More explanations have been included in the revision.

Reviewer #3 (Comments to the Authors (Required)):

Yeh et al, J Cell Bio 202306117

on this manuscript Yeh and colleagues use a combination of genetic, biochemical, fluorescence and structural approaches to dissect the structure and function of the Bcp1, Rkm1 and uL14 complex. The experiments are well executed and mostly support the conclusions on the paper. The structure of the complex is not up to current standard in the field however, and overall, it is a borderline candidate for JCB as its relevance for a broader audience is somewhat tenuous.

Major comments:

- It is not true that the ternary complex of Bcp1, Rkm1 and uL14 is too small for cryo-EM. The complex is over 120 kDa, well within the range of current cryo-EM capabilities. As such the cryo-EM data are not current standard. The data claims a resolution of 5.9Å, but the map looks significantly worse, and there is no FSC curve, and it is not noted which criteria for this resolution was used.

Sorry for the confusion. What we meant is the smaller size and structural heterogeneity prevent the high-resolution cryo-EM structure determination, not simply the size issue. To avoid the confusion, we removed the smaller size in the revised manuscript. We collected data using 300kV Titan Krios equipped with a K3 detector. A new data was collected using purified cross-linked Bcp1/uL14/Rkm1 complex. Unfortunately, we still failed to improve the map resolution.

In the original manuscript, we purposely removed this FSC-overestimated 5.9 Å resolution from the main text but mistakenly left it in the figure. Therefore, there is no FSC curve. As we explained to Reviewer 1, we now use the d99 approach to estimate the resolution based on a real space map and obtain the resolution of 16.32 Å.

- The authors must either obtain a much better-quality structure or probe the suggested placement of residues more extensively by making mutants at each of the protein-protein interfaces. These will then also be useful in functional studies (see below).

We have tried to collect another dataset on Titan Krios but failed to obtain a high-resolution cryo-EM structure. As we showed in the Fig 4G and supplementary movie 1, the trimer can be dynamic, preventing better-quality structure. The dynamics nature shows multiple interfaces all partially participate in the interactions, presenting a challenging case to create suitable mutants. However, we could only show that uL14(Δ loop) lost interaction with Bcp1, consistent with our docking model.

Having said that, our docking model, guided by experimental cross-linking data, reveals the spatial correlation between Bcp1, Rkm1, and uL14 and demonstrates the protection model. The limited proteolysis experiments (shown in the revised figure 1F) suggested by Reviewer 2 support our spatial model that Bcp1 and Rkm1 protect different regions of uL14.

- Figure 2 needs an input as well as the elution. Without these it is impossible to judge how well this experiment works.

The input has been included in the revision.

- If Bcp1 really dissociates the Rkm1/uL14 complex, then one would expect that a Bcp1 mutant that fails to interact with uL14 and Rkm1 should be deleterious, and that its defect should be suppressable by a mutation in Rkm1 that weakens the binding to uL14

Bcp1 serves not only to dissociate uL14 from Rkm1 but also from Kap. Its central and indispensable function resides in facilitating the dissociation of the Rkm1/uL14 complex from Kap, as the transport pathway of uL14 is vital, whereas the

methyations of uL14 are not. Thus, to find such mutants proposed here might be not easy.

The role of Bcp1 in dissociating uL14 from Kap has been published in the previous study (Ting et al, 2017). To further substantiate the role of Bcp1 in dissociating the Rkm1/uL14 complex from Kap, we conducted several experiments. Bcp1 could dissociate the Rkm1/uL14 complex from Kap in vitro (Fig 2E). The loss of uL14's interaction with Bcp1 (uL14 Δ loop) led to the accumulation of Rkm1 on Kap (Fig 5G).

We also demonstrate that Bcp1 is required to release uL14 from Rkm1. Immuno-purification of uL14 and uL14(RR) revealed that uL14(RR) exhibited heightened signals for Bcp1 and Rkm1 interactions (Fig 6G). Similarly, rkm1(Y273F) showed stronger interactions with Bcp1 and uL14 (Fig 6F). These findings demonstrate that uL14(RR), with mutations at the methylation sites, and rkm1(Y273F), with mutations at the methyltransferase site, were retained in the form of a ternary complex. Additionally, overexpression of uL14(RR) (Fig 6I) and rkm1(Y273F) (Fig 6H) was toxic to the growth of *bcp1ts*.

The provided data strongly supports the proposed model depicting Bcp1's role in dissociating uL14 from both Kap and Rkm1.

Ting, Y.H., T.J. Lu, A.W. Johnson, J.T. Shie, B.R. Chen, S.S. Kumar, and K.Y. Lo. 2017. Bcp1 is the Nuclear Chaperone of the 60S ribosomal protein Rpl23 in *Saccharomyces cerevisiae*. *J Biol Chem*. 292:585-596.

- The IF data do not really support the model for how the process works. The data indicate that Rkm1 depends on uL14 for nuclear import, and that Bcp1 depends on binding Kap for nuclear import. This latter piece of data is not consistent with the model that Bcp1 releases uL14 (or the uL14 Rkm1 complex) from karyopherins in the nucleus.

Bcp1 is a shuttling protein (Audhya et al, 2003) and relied on a karyopherin for its import. It's important to note that the import processes of Bcp1 and uL14 are independent events. Bcp1 serves as the escortin and chaperone for uL14 in the nucleus (Ting et al, 2017). We provide a more detailed and clear explanation of this aspect in the revised manuscript.

Audhya, A., and S.D. Emr. 2003. Regulation of PI4,5P2 synthesis by nuclear-cytoplasmic shuttling of the Mss4 lipid kinase. *EMBO J*. 22:4223-4236.

Ting, Y.H., T.J. Lu, A.W. Johnson, J.T. Shie, B.R. Chen, S.S. Kumar, and K.Y. Lo. 2017. Bcp1 is the Nuclear Chaperone of the 60S ribosomal protein Rpl23 in *Saccharomyces cerevisiae*. *J Biol Chem.* 292:585-596.

- The finding that Bcp1-dependent release of Rkm1 from uL14 requires its methylation is very interesting. However, it is NOT supportive of the conclusion that this is a QC step for uL14 functionality. It merely says that it is a QC step for uL14 methylation. To show that it monitors uL14 integrity, the authors would need to show that a misfolded mutant cannot be released. I will let the authors and the editor decide whether it is appropriate to just tone down the conclusion or carry out this other experiment.

Thanks for the suggestion. We included uL14 Δ loop, which contains methylation sites but loses the contact with Bcp1, in the methylation assay (Fig 6E). Although this mutant maintains the association with Rkm1, it can't be methylated and released by Bcp1. The data suggest if uL14 is in incorrect interaction with Rkm1 and Bcp1 would be retained from loading to the ribosome. Thus, we propose that this is a QC step for the uL14.

Minor comments:

- There are 46 proteins in the yeast 60S subunit, not 39 as claimed in the introduction.
- The manuscript could benefit from some English language editing
- P.13 has uL14 as ul14.

Thanks for the suggestions. We have revised accordingly in the revision.

- Figures 1 and 2 refer to uL14 alternately as uL14 or L23. The authors need to pick one nomenclature.

The gene name of uL14 is *RPL23A* and *RPL23B* in yeast. Thus, the strain we used is the *RPL23A* gene under the *GAL*-driven promoter and labelled as *GAL::RPL23*. We included a sentence for description in the revision.

March 11, 2024

RE: JCB Manuscript #202306117R-A

Prof. Kai-Yin Lo
National Taiwan University
No. 1, Sec. 4, Roosevelt Rd.
Taipei 10617
Taiwan

Dear Prof. Lo:

Thank you for submitting your revised manuscript entitled "The dual protection system for ribosomal protein uL14 by its dedicated chaperone and methyltransferase." We would be happy to publish your paper in JCB pending final revisions necessary to meet our formatting guidelines (see details below).

A. MANUSCRIPT ORGANIZATION AND FORMATTING:

1) Text limits: Character count for Articles is < 40,000, not including spaces. Count includes title page, abstract, introduction, results, discussion, and acknowledgments. Count does not include materials and methods, figure legends, references, tables, or supplemental legends.

Please also consider Reviewer #2 comment about text organization as you prepare your final files.

2) Figure formatting: Articles may have up to 10 main text figures. Scale bars must be present on all microscopy images, including inset magnifications. Molecular weight or nucleic acid size markers must be included on all gel electrophoresis - please add MW labels to Figures 1B/C/D/E/F, 2B/E/F, 4D/E, 5E/F/G, 6A/B/D/E/F/G, & S1B/D/E.

Also, please avoid pairing red and green for images and graphs to ensure legibility for color-blind readers. If red and green are paired for images, please ensure that the particular red and green hues used in micrographs are distinctive with any of the colorblind types. If not, please modify colors accordingly or provide separate images of the individual channels.

3) Statistical analysis: Error bars on graphic representations of numerical data must be clearly described in the figure legend. The number of independent data points (n) represented in a graph must be indicated in the legend. Please, indicate whether 'n' refers to technical or biological replicates (i.e. number of analyzed cells, samples or animals, number of independent experiments). If independent experiments with multiple biological replicates have been performed, we recommend using distribution-reproducibility SuperPlots (please see Lord et al., JCB 2020) to better display the distribution of the entire dataset, and report statistics (such as means, error bars, and P values) that address the reproducibility of the findings.

Statistical methods should be explained in full in the materials and methods. For figures presenting pooled data the statistical measure should be defined in the figure legends. Please also be sure to indicate the statistical tests used in each of your experiments (both in the figure legend itself and in a separate methods section) as well as the parameters of the test (for example, if you ran a t-test, please indicate if it was one- or two-sided, etc.). Also, if you used parametric tests, please indicate if the data distribution was tested for normality (and if so, how). If not, you must state something to the effect that "Data distribution was assumed to be normal but this was not formally tested."

4) Abstract and title: Please make the corrections in the abstract recommended by Reviewer #2. To convey the advance more clearly, we also suggest revising the title to: "Dual protection by Bcp1 and Rkm1 ensures incorporation of uL14 into pre-60S ribosomal subunits"

5) Materials and methods: Should be comprehensive and not simply reference a previous publication for details on how an experiment was performed. Please provide full descriptions (at least in brief) in the text for readers who may not have access to referenced manuscripts. The text should not refer to methods "...as previously described." Please also indicate the type of membrane used for immunoblotting as well as describe acquisition and quantification methods.

6) For all cell lines, vectors, constructs/cDNAs, etc. - all genetic material: please include database / vendor ID (e.g., Addgene, ATCC, etc.) or if unavailable, please briefly describe their basic genetic features, even if described in other published work or

gifted to you by other investigators (and provide references where appropriate). Please be sure to provide the sequences for all of your oligos: primers, si/shRNA, RNAi, gRNAs, etc. in the materials and methods. You must also indicate in the methods the source, species, and catalog numbers/vendor identifiers (where appropriate) for all of your antibodies, including secondary. If antibodies are not commercial, please add a reference citation if possible.

7) Microscope image acquisition: The following information must be provided about the acquisition and processing of images:

- a. Make and model of microscope
- b. Type, magnification, and numerical aperture of the objective lenses
- c. Temperature
- d. Imaging medium
- e. Fluorochromes
- f. Camera make and model
- g. Acquisition software
- h. Any software used for image processing subsequent to data acquisition. Please include details and types of operations involved (e.g., type of deconvolution, 3D reconstitutions, surface or volume rendering, gamma adjustments, etc.).

8) References: There is no limit to the number of references cited in a manuscript. References should be cited parenthetically in the text by author and year of publication. Abbreviate the names of journals according to PubMed.

9) Supplemental materials: Articles generally have 5 supplemental figures and up to 10 videos. Figures cannot span more than a single page. For Figures S3 & S5 which are multiple pages each, please consolidate the panels as much as possible. If you cannot consolidate all panels into a single page then split into separate figures. We will be able to give you some extra space. Please also note that tables, like figures, should be provided as individual, editable files. A summary of all supplemental material should appear at the end of the Materials and methods section. Please include one brief sentence per item.

10) Video legends: Should describe what is being shown, the cell type or tissue being viewed (including relevant cell treatments, concentration and duration, or transfection), the imaging method (e.g., time-lapse epifluorescence microscopy), what each color represents, how often frames were collected, the frames/second display rate, and the number of any figure that has related video stills or images.

11) eTOC summary: A ~40-50 word summary that describes the context and significance of the findings for a general readership should be included on the title page. The statement should be written in the present tense and refer to the work in the third person. It should begin with "First author name(s) et al..." to match our preferred style.

13) A separate author contribution section is required following the Acknowledgments in all research manuscripts. All authors should be mentioned and designated by their first and middle initials and full surnames. We encourage use of the CRediT nomenclature (<https://casrai.org/credit/>).

14) ORCID IDs: ORCID IDs are unique identifiers allowing researchers to create a record of their various scholarly contributions in a single place. Please note that ORCID IDs are required for all authors. At resubmission of your final files, please be sure to provide your ORCID ID and those of all co-authors.

15) JCB requires authors to submit Source Data used to generate figures containing gels and Western blots with all revised manuscripts. This Source Data consists of fully uncropped and unprocessed images for each gel/blot displayed in the main and supplemental figures. Since your paper includes cropped gel and/or blot images, please be sure to provide one Source Data file for each figure that contains gels and/or blots along with your revised manuscript files. File names for Source Data figures should be alphanumeric without any spaces or special characters (i.e., SourceDataF#, where F# refers to the associated main figure number or SourceDataFS# for those associated with Supplementary figures). The lanes of the gels/blots should be labeled as they are in the associated figure, the place where cropping was applied should be marked (with a box), and molecular weight/size standards should be labeled on all images. Source Data files will be directly linked to specific figures in the published article.

15) Journal of Cell Biology now requires a data availability statement for all research article submissions. These statements will be published in the article directly above the Acknowledgments. The statement should address all data underlying the research presented in the manuscript. Please visit the JCB instructions for authors for guidelines and examples of statements at

(<https://rupress.org/jcb/pages/editorial-policies#data-availability-statement>).

B. FINAL FILES:

Thank you for your attention to these final processing requirements. Please revise and format the manuscript and upload materials within 7-10 days. If you need an extension for whatever reason, please let us know and we can work with you to determine a suitable revision period.

Thank you for this interesting contribution, we look forward to publishing your paper in Journal of Cell Biology.

Sincerely,

Michael Rout, PhD
Monitoring Editor
Journal of Cell Biology

Dan Simon, PhD
Scientific Editor
Journal of Cell Biology

Reviewer #2 (Comments to the Authors (Required)):

The authors have adequately addressed most of my concerns about scientific issues.

However, I still believe that additional editing for English and grammar is necessary. There seem to be numerous small errors. One frequent mistake: "data" should be treated as a plural noun.

Another example: I have made the following small changes to the Abstract: additions in CAPS following appropriate deletions.

Abstract

17 Eukaryotic ribosomal proteins contain extended regions essential for translation
18 coordination. Dedicated chaperones stabilize the associated ribosomal proteins. We
19 identified Bcp1 as the chaperone of uL14 in *Saccharomyces cerevisiae*. Rkm1, the
20 lysine methyltransferase of uL14, forms a ternary complex with Bcp1 and uL14.... in
21 protection of...TO PROTECT... uL14. Rkm1 is transported with uL14 by importins to the nucleus, and

22 Bcp1 disassembles Rkm1 and importin from uL14 simultaneously in a
23 RanGTP-independent manner. Molecular docking, guided by cross-linking mass
24 spectrometry and validated by a low-resolution cryo-EM map, reveals the correlation
25 between Bcp1, Rkm1, and uL14, demonstrating the protection model. In addition, the
26 ternary complex also serves as a surveillance point, whereas incorrect uL14 is
27 retained on the...DELETE "THE"?... Rkm1 and prevented from loading to the pre-60S... RIBOSOMAL SUBUNITS. This study
reveals
28 the molecular mechanism of how uL14 is protected and quality checked by serial
29 steps to ensure the...ITS... safe delivery from ...THE...cytoplasm till...UNTIL... until its incorporation into DELETE
"THE"...the... 60S RIBOSOMAL SUBUNITS.

To improve the clarity of the description of the rationale for this work, I suggest moving the first paragraph in the Discussion, describing ribosomal proteins and their chaperones, to the second paragraph of the Introduction. In addition, maybe move the current third paragraph of the Introduction to after the current fourth paragraph of the Introduction.

Reviewer #3 (Comments to the Authors (Required)):

this revised manuscript addresses my previous concerns adequately.
I still think mutants would have been nice.